# A SIMPLE CONNECTION FROM LOSS FLATNESS TO COMPRESSED REPRESENTATIONS IN NEURAL NETWORKS

## ABSTRACT

The generalization capacity of deep neural networks has been studied in a variety of ways, including at least two distinct categories of approaches: one based on the shape of the loss landscape in parameter space, and the other based on the structure of the representation manifold in feature space (that is, in the space of unit activities). Although these two approaches are related, they are rarely studied together explicitly. Here, we present an analysis that bridges this gap. We show that in the final phase of learning in deep neural networks, the compression of the manifold of neural representations correlates with the flatness of the loss around the minima explored by SGD. This correlation is predicted by a relatively simple mathematical relationship: a flatter loss corresponds to a lower upper bound on the compression metrics of neural representations. Our work builds upon the linear stability insight by Ma and Ying, deriving inequalities between various compression metrics and quantities involving sharpness. Empirically, our derived inequality predicts a consistently positive correlation between representation compression and loss sharpness in multiple experimental settings. Overall, we advance a dual perspective on generalization in neural networks in both parameter and feature space.

## 1 INTRODUCTION

Deep neural networks' generalization capacity has been studied in many ways. Generalization is a complex phenomenon influenced by myriad factors, including model architecture, dataset size and diversity, and the specific task used to train a network. Researchers continue to develop new techniques to enhance generalization (Elsayed et al., 2018; Galanti et al., 2023). From a theoretical point of view, we can identify two distinct categories of approach. These are works that study neural network generalization in the context of (a) properties of minima of the loss function that learning algorithms find in parameter space (Dinh et al., 2017; Andriushchenko et al., 2023), and (b) properties of the representations that optimized networks find in feature space – that is, in the space of their neural activation (Rangamani et al., 2023; Papyan et al., 2020).

One of the most widely studied factors that influence generalization is the shape of the loss landscape in parameter space. Empirical studies and theoretical analyses have shown that training deep neural networks using stochastic gradient descent (SGD) with a small batch size and a large learning rate often converges to flat and wide minima (Ma & Ying, 2021; Blanc et al., 2020; Geiger et al., 2021; Li et al., 2022; Wu et al., 2018b; Jastrzebski et al., 2018; Xie et al., 2021; Zhu et al., 2019). Flat minima refer to regions in the loss landscape where the loss function has a relatively large basin: put simply, the loss doesn't change much in different directions around the minimum. Many works conjecture that flat minima lead to a simpler model (shorter description length), and thus are less likely to overfit and more likely to generalize well (Jastrzebski et al., 2018; Yang et al., 2023; Wu et al., 2018b). However, whether flatness positively correlates with the network's generalization capability remains unsettled (Dinh et al., 2017; Andriushchenko et al., 2023; Yang et al., 2021). In particular, Dinh et al. (2017) argues that one can construct very sharp networks that generalize well through reparametrization. However, more recent work (Andriushchenko et al., 2023) shows that even reparametrization-invariant sharpness cannot capture the relationship between sharpness and generalization.

In our work, we investigate how the sharpness of the loss function near learned solutions in parameter space influences local geometric features of neural representations. We demonstrate that as this sharpness decreases and the minima become flatter, there is a set of mathematical bounds that imply that the neural representation must undergo at least a specific, computable level of compression. This process, which is related to previous results including the concept of neural collapse (Farrell et al., 2022; Kothapalli et al., 2022; Zhu et al., 2021; Ansuini et al., 2019; Recanatesi et al., 2019; Papyan et al., 2020), refers to the emergence of a more compact and by some measures lower dimensional structure in the neural representation space. Compression in the feature space enables networks to isolate the most crucial and discriminative features of input data. As a model becomes less sensitive to small perturbations or noise in the input data, it gains increased robustness against variations between training and test data. This simple and direct relationship between compression and robustness creates a valuable lens into networks' potential to generalize.

We find that bounds that apply to two different metrics of compression – volumetric ratio and maximum local sensitivity – include different terms, and therefore predict different levels of compression for each. We also note that local dimensionality is a compression metric of a distinct nature, and therefore does not necessarily correlate with sharpness. Taken together, this reveals that the impact of loss function sharpness on the neural representation is more complex than a simple (and single) compression effect. These effects shed light on the complex link between sharpness and generalization.

Throughout, we focus on the second, or final, stage of learning, which proceeds after SGD has already found parameters that give near-optimal performance (i.e., zero training *error*) on the training data (Ma & Ying, 2021; Tishby & Zaslavsky, 2015; Ratzon et al., 2023). Here, additional learning still occurs, which changes the properties of the solutions in both feature and parameter space in very interesting ways.

Our work makes the following novel contributions:

1. The paper identifies two representation space quantities that quantify compression and are bounded by sharpness – volumetric ratio and maximum local sensitivity (MLS) – and gives new explicit formulas for these bounds.
2. The paper conducts empirical experiments with VGG-11, LeNet, MLP, and ViT networks and finds that volume compression and MLS are indeed strongly correlated with sharpness.
3. The paper finds that only the bound (Proposition 3.10) incorporating all linear weights of the network consistently predicts a positive correlation between both sides of the inequality across various experimental settings.

In these ways, we help reveal the interplay between key properties of trained neural networks in parameter space and representation space. Specifically, we identify a sequence of inequality conditions for the bounds that link the volume and MLS of the neural representations to the sharpness in parameter space. These conditions help explain why there are mixed results on the relationship between sharpness and generalization in the literature, by looking through the additional lens of the induced representations. Our findings altogether suggest that allied views into representation space offer a valuable dual perspective to that of parameter space landscapes for understanding the effects of learning on generalization.

Our paper proceeds as follows. First, we review arguments of Ma & Ying (2021) that flatter minima can constrain the gradient of the loss with respect to network inputs and extend the formulation to the multidimensional input case (Section 2). Next, we prove that lower sharpness implies a lower upper bound on two metrics of the compression of the representation manifold in feature space: the local volume and the maximum local sensitivity (MLS) (Section 3.1, Section 3.2). We conclude our findings with simulations that confirm our central theoretical results and show how they can be applied in practice (Section 4).

## 2 BACKGROUND AND SETUP

Consider a feedforward neural network $f$ with input data $\mathbf{x} \in \mathbb{R}^M$ and parameters $\boldsymbol{\theta}$. The output of the network is:
$$\mathbf{y} = f(\mathbf{x}; \boldsymbol{\theta}) , \tag{1}$$

where $\mathbf{y} \in \mathbb{R}^N$ ($N < M$). We consider a quadratic loss $L(\mathbf{y}, \mathbf{y}_{\text{true}}) = \frac{1}{2}||\mathbf{y} - \mathbf{y}_{\text{true}}||^2$, a function of the outputs and ground truth $\mathbf{y}_{\text{true}}$. In the following, we will simply write $L(\mathbf{y})$, $L(f(\mathbf{x}, \boldsymbol{\theta}))$ or simply $L(\boldsymbol{\theta})$ to highlight the dependence of the loss on the output, the network or its parameters. We note that for the cross-entropy loss function, the Hessian vanishes as the cross-entropy loss approaches 0 (Granziol, 2020; Wu et al., 2018a). Therefore, the sharpness of cross-entropy loss cannot differentiate between local minima with different traces of the Hessian. As a result, (Granziol, 2020) showed that SGD may find a minimum with lower loss, hence flatter, but the minimum is overfitting, leading to worse generalization performance. However, our result readily extends to logistic loss with label smoothing (ref. Lemma A.13 in (Wen et al., 2023)).

During the last phase of learning, Ma and colleagues have recently argued that SGD appears to regularize the sharpness of the loss (Li et al., 2022) (see also (Wu et al., 2018b; Jastrzebski et al., 2018; Xie et al., 2021; Zhu et al., 2019)). This means that the dynamics of SGD lead network parameters to minima where the local loss landscape is flatter or wider. This is best captured by the sharpness, measured by the sum of the eigenvalues of the Hessian:

$$S(\boldsymbol{\theta}) = \text{Tr}(H) , \tag{2}$$

with $H = \nabla^2 L(\boldsymbol{\theta})$ being the Hessian. A solution with low sharpness is a flatter solution. Following (Ma & Ying, 2021; Ratzon et al., 2023), we define $\boldsymbol{\theta}^*$ to be an "exact interpolation solution" on the zero training loss manifold in the parameter space (the zero loss manifold in what follows), where $f(\mathbf{x}_i, \boldsymbol{\theta}^*) = \mathbf{y}_i$ for all $i$'s (with $i \in \{1..n\}$ indexing the training set) and $L(\boldsymbol{\theta}^*) = 0$. On the zero loss manifold, in particular, we have

$$S(\boldsymbol{\theta}^*) = \frac{1}{n} \sum_{i=1}^{n} \|\nabla_{\boldsymbol{\theta}} f(\mathbf{x}_i, \boldsymbol{\theta}^*)\|_F^2, \tag{3}$$

where $\|\cdot\|_F$ is the Frobenius norm. We state a proof of this equality, which appears in Ma & Ying (2021) and Wen et al. (2023), in Appendix B. In practice, the parameter $\boldsymbol{\theta}$ will never reach an exact interpolation solution due to the gradient noise of SGD; however, Equation (3) is a good enough approximation of the sharpness as long as we find an approximate interpolation solution (see error bounds in our Lemma. B.1).

To see why minimizing the sharpness of the solution leads to more compressed representations, we need to move from the parameter space to the input space. To do so we review a pioneering argument of Ma & Ying (2021) that relates variations in the input data $\mathbf{x}$ and input weights. Let $\mathbf{W}$ be the input weights (the parameters of the first linear layer) of the network, and $\bar{\boldsymbol{\theta}}$ be the rest of the parameters. Following (Ma & Ying, 2021), as the weights $\mathbf{W}$ multiply the inputs $\mathbf{x}$, we have the following identities:

$$\|\nabla_{\mathbf{W}} f(\mathbf{W}\mathbf{x}; \bar{\boldsymbol{\theta}})\|_F = \sqrt{\sum_{i,j,k} J_{jk}^2 x_i^2} = \|J\|_F \|\mathbf{x}\|_2 \geq \|J\|_2 \|\mathbf{x}\|_2 ,$$

$$\nabla_{\mathbf{x}} f(\mathbf{W}\mathbf{x}; \bar{\boldsymbol{\theta}}) = J\mathbf{W} , \tag{4}$$

where $J = \frac{\partial f(\mathbf{W}\mathbf{x}; \bar{\boldsymbol{\theta}})}{\partial(\mathbf{W}\mathbf{x})}$ is a complex expression as computed in, for example, backpropagation. From Equation (4) and the sub-multiplicative property of the Frobenius norm and the matrix 2-norm [1], we have:

$$\|\nabla_{\mathbf{x}} f(\mathbf{W}\mathbf{x}; \bar{\boldsymbol{\theta}})\|_2 \leq \|\nabla_{\mathbf{x}} f(\mathbf{W}\mathbf{x}; \bar{\boldsymbol{\theta}})\|_F \leq \frac{\|\mathbf{W}\|_2}{\|\mathbf{x}\|_2} \|\nabla_{\mathbf{W}} f(\mathbf{W}\mathbf{x}; \bar{\boldsymbol{\theta}})\|_F . \tag{5}$$

We call Equation (5) the linear stability trick. This in turn reveals the impact of flatness (when $k = 2$) on the input sensitivity:

$$\frac{1}{n} \sum_{i=1}^{n} \|\nabla_{\mathbf{x}} f(\mathbf{x}_i, \boldsymbol{\theta}^*)\|_2^k \leq \frac{1}{n} \sum_{i=1}^{n} \|\nabla_{\mathbf{x}} f(\mathbf{x}_i, \boldsymbol{\theta}^*)\|_F^k \leq \frac{\|\mathbf{W}\|_2^k}{\min_i \|\mathbf{x}_i\|_2^k} \frac{1}{n} \sum_{i=1}^{n} \|\nabla_{\mathbf{W}} f(\mathbf{x}_i, \boldsymbol{\theta}^*)\|_F^k$$

$$\leq \frac{\|\mathbf{W}\|_2^k}{\min_i \|\mathbf{x}_i\|_2^k} \frac{1}{n} \sum_{i=1}^{n} \|\nabla_{\boldsymbol{\theta}} f(\mathbf{x}_i, \boldsymbol{\theta}^*)\|_F^k . \tag{6}$$

---

[1] $\|AB\|_F \leq \|A\|_F \|B\|_2$, $\|AB\|_2 \leq \|A\|_2 \|B\|_2$

Thus, the effect of input perturbations is constrained by the sharpness of the loss function (compare with Equation equation 3). The flatter the minimum of the loss, the lower the effect of input space perturbations on the network function $f(\mathbf{x}, \boldsymbol{\theta}^*)$ as determined by gradients.

While the experiments of Ma & Ying (2021) show a high correlation between the left-hand side of Equation (6) and the sharpness, Equation (6) does not explain such a correlation by itself because of the scaling factor $\|\mathbf{W}\|_2^k / \min_i \|\mathbf{x}_i\|_2^k$. This factor makes the right-hand side of Equation (6) highly variable, leading to mixed positive and/or negative correlations with sharpness under different experimental settings. In the next section, we will improve this bound to relate sharpness to various metrics measuring robustness and compression of representations. More specifically, compared to formula (4) of Ma & Ying (2021), we make the following improvements:

1. We replace the minimum with the geometric mean to achieve a more stable bound. This term remains relevant as common practice in deep learning does *not* normalize the input by its 2-norm, as this would erase information about the modulus of the input.

2. While Ma & Ying (2021) only considers scalar output, we extend the result to networks with multi-dimensional input and output.

3. We introduce new metrics such as Network Volumetric Ratio and Network MLS (Definition 3.5 and Definition 3.9) and their sharpness-related bounds, which have two advantages compared prior results (cf. the right-hand side of Equation (6)): 1) our metrics consider all linear weights, so that bounds remain stable to weight changes during training . 2) they avoid the gap between derivative w.r.t. the first layer weights and the derivative w.r.t. all weights, i.e. the second inequality in Eq. 6, thus tightening the bound.

Moreover, we show that the underlying theory readily extends to networks with residual connections in Appendix A.

## 3 FROM ROBUSTNESS TO INPUTS TO COMPRESSION OF REPRESENTATIONS

We now further analyze variations in the input and how they propagate through the network to shape representations of sets of inputs. Overall, we focus on three key metrics of network representations: local dimensionality, volumetric ratio, and maximum local sensitivity. These quantities enable us to establish and evaluate the influence of input variations and, in turn, sharpness on neural representation properties.

### 3.1 WHY SHARPNESS BOUNDS LOCAL VOLUMETRIC TRANSFORMATION IN REPRESENTATION SPACE

Now we quantify how a network compresses its input volumes via the local volumetric ratio, between a hypercube of side length $h$ at $\mathbf{x}$ and its image under transformation $f$:

$$
\begin{aligned}
d\,\mathrm{Vol}|_{f(\mathbf{x}, \boldsymbol{\theta}^*)} &= \lim_{h \to 0} \frac{\mathrm{Vol}(f(\mathbf{x}, \boldsymbol{\theta}^*))}{\mathrm{Vol}(\mathbf{x})} \\
&= \sqrt{\det\left(\nabla_{\mathbf{x}} f^T \nabla_{\mathbf{x}} f\right)} ,
\end{aligned}
\tag{7}
$$

which is equal to the square root of the product of all positive eigenvalues of $C_f^{\mathrm{lim}}$.

**Definition 3.1.** *The **Local Volumetric Ratio** of a network $f$ with parameter $\boldsymbol{\theta}$ at input $\mathbf{x}$ is defined as $d\,\mathrm{Vol}|_{f(\mathbf{x}, \boldsymbol{\theta})} = \sqrt{\det\left(\nabla_{\mathbf{x}} f^T \nabla_{\mathbf{x}} f\right)}$.*

Exploiting the bound on the gradients derived earlier in Equation (5), we derive a similar bound for the volumetric ratio:

**Lemma 3.2.**

$$
d\,\mathrm{Vol}|_{f(\mathbf{x}, \boldsymbol{\theta}^*)} \le \left(\frac{\mathrm{Tr}\,\nabla_{\mathbf{x}} f^T \nabla_{\mathbf{x}} f}{N}\right)^{N/2} = N^{-N/2} \|\nabla_{\mathbf{x}} f(\mathbf{x}, \boldsymbol{\theta}^*)\|_F^N ,
\tag{8}
$$

where the first inequality uses the inequality of arithmetic and geometric means and the second the definition of the Frobenius norm. Next we introduce a measure of the volumetric ratio averaged across input samples.

**Definition 3.3.** *The **Mean Local Volumetric Ratio** of a network $f$ with parameter $\boldsymbol{\theta}$ is defined as the sample mean of Local Volumetric Ratio: $dV_{f(\boldsymbol{\theta})} = \frac{1}{n}\sum_{i=1}^{n} d\operatorname{Vol}|_{f(\mathbf{x}_i,\boldsymbol{\theta})}$.*

Then we have the following inequality that relates the sharpness to the mean local volumetric ratio:

**Proposition 3.4.** *The local volumetric ratio is upper bounded by a sharpness related quantity:*

$$dV_{f(\boldsymbol{\theta}^*)} \le \frac{N^{-N/2}}{n}\sum_{i=1}^{n}\|\nabla_{\mathbf{x}}f(\mathbf{x},\boldsymbol{\theta}^*)\|_F^N \le \frac{1}{n}\sqrt{\sum_{i=1}^{n}\frac{\|\mathbf{W}\|_2^{2N}}{\|\mathbf{x}_i\|_2^{2N}}\left(\frac{nS(\boldsymbol{\theta}^*)}{N}\right)^{N/2}} \qquad (9)$$

*for all $N \ge 1$.*

The proof of the above inequalities is given in Appendix C. Next, we give an inequality that is obtained by applying Equation (9) to every intermediate layer. Instead of only considering the input layer, all linear weights (including any convolution layers) are taken into account. Denote the input to the $l$-th linear layer as $\mathbf{x}_i^l$ for $l = 1, 2, \cdots, L$. In particular, $\mathbf{x}_i^1 = \mathbf{x}_i$ is the input of the entire network. Similarly, $\mathbf{W}_l$ is the weight matrix of $l$-th linear/convolutional layer. With a slight abuse of notation, we use $f_l$ to denote the mapping from the input of the $l$-th layer to the final output. Then we define network volumetric ratio:

**Definition 3.5.** *The **Network Volumetric Ratio** is defined as the sum of mean local volumetric ratio $dV_{f_l}$ for all $l$, i.e. $dV_{net} = \sum_{l=1}^{L} dV_{f_l}$*

Then we have the following inequality:

**Proposition 3.6.** *The network volumetric ratio is upper bounded by a sharpness related quantity:*

$$\sum_{l=1}^{L} dV_{f_l} \le \frac{N^{-N/2}}{n}\sum_{l=1}^{L}\sum_{i=1}^{n}\|\nabla_{\mathbf{x}^l}f_i^l\|_F^N \le \frac{1}{n}\sqrt{\sum_{l=1}^{L}\sum_{i=1}^{n}\frac{\|\mathbf{W}_l\|_2^{2N}}{\|\mathbf{x}_i^l\|_2^{2N}} \cdot \left(\frac{nS(\boldsymbol{\theta}^*)}{N}\right)^{N/2}}. \qquad (10)$$

Again a detailed derivation of the above inequalities is given in Appendix C. Proposition 3.4 and Proposition 3.6 imply that flatter minima of the loss function in parameter space contribute to the compression of the data's representation manifold.

### 3.2 MAXIMUM LOCAL SENSITIVITY AS AN ALLIED METRIC TO TRACK NEURAL REPRESENTATION GEOMETRY

We observe that the equality condition in the first line of Equation (8) rarely holds in practice. To achieve equality, we would need all singular values of the Jacobian matrix $\nabla_{\mathbf{x}}f$ to be identical. However, our experiments in Section 4 show that the local dimensionality decreases rapidly with training onset; this implies that $\nabla_{\mathbf{x}}f^T\nabla_{\mathbf{x}}f$ has a non-uniform eigenspectrum (i.e., some directions being particularly elongated, corresponding to a lower overall dimension). Moreover, the volume will decrease rapidly as the smallest eigenvalue vanishes. Thus, although sharpness upper bounds the volumetric ratio and often correlates reasonably with it (see experiments in Appendix G.2), the correlation is far from perfect.

Fortunately, considering only the maximum eigenvalue instead of the product of all eigenvalues alleviates this discrepancy (recall that $\det\left(\nabla_{\mathbf{x}}f^T\nabla_{\mathbf{x}}f\right)$ in Equation (7) or volumetric ratio definition is the product of all eigenvalues).

**Definition 3.7.** *The **Maximum Local Sensitivity (MLS)** of network $f$ is defined to be $\operatorname{MLS}_f = \frac{1}{n}\sum_{i=1}^{n}\|\nabla_{\mathbf{x}}f(\mathbf{x}_i)\|_2$, which is the sample mean of the largest singular value of $\nabla_{\mathbf{x}}f$.*

Intuitively, MLS is the largest possible average local change of $f(\mathbf{x})$ when the norm of the perturbation to $\mathbf{x}$ is regularized. Given this definition, we can obtain a bound on MLS below.

**Proposition 3.8.** *The maximum local sensitivity is upper bounded by a sharpness related quantity:*

$$\operatorname{MLS}_f = \frac{1}{n}\sum_{i=1}^{n}\|\nabla_{\mathbf{x}}f(\mathbf{x}_i,\boldsymbol{\theta}^*)\|_2 \le \|\mathbf{W}\|_2\sqrt{\frac{1}{n}\sum_{i=1}^{n}\frac{1}{\|\mathbf{x}_i\|_2^2}}S(\boldsymbol{\theta}^*)^{1/2}. \qquad (11)$$

The derivation of the above bound is included in Appendix D. As an alternative measure of compressed representations, we empirically show in Appendix G.2 that MLS has a higher correlation with sharpness and test loss than local volumetric ratio. We include more analysis of the tightness of this bound in Appendix G and discuss its connection to other works therein.

Similar to the network volumetric ratio, a straightforward extension of MLS is the Network MLS (NMLS), which we define as the average of MLS w.r.t. input to each linear layer.

**Definition 3.9.** *The **Network Maximum Local Sensitivity (NMLS)** of network $f$ is defined as the sum of* $\mathrm{MLS}_{f_l}$ *for all l, i.e.* $\sum_{l=1}^{L} \mathrm{MLS}_{f_l}$.

Recall that $\mathbf{x}_i^l$ is the input to the $l$-th linear/convoluational layer for sample $\mathbf{x}_i$ and $f_l$ is the mapping from the input of $l$-th layer to the final output. Again we have the following inequality:

**Proposition 3.10.** *The network maximum local sensitivity is upper bounded by a sharpness related quantity:*

$$\mathrm{NMLS} = \frac{1}{n} \sum_{l=1}^{L} \sum_{i=1}^{n} \|\nabla_{\mathbf{x}^l} f^l(\mathbf{x}_i^l, \boldsymbol{\theta}^*)\|_2 \le \sqrt{\frac{1}{n} \sum_{i=1}^{n} \sum_{l=1}^{L} \frac{\|\mathbf{W}_l\|_2^2}{\|\mathbf{x}_i^l\|^2} \cdot S(\boldsymbol{\theta}^*)^{1/2}}. \tag{12}$$

The derivation is in Appendix D. The advantage of NMLS is that instead of only considering the robustness of the final output w.r.t. the input, NMLS considers the robustness of the output w.r.t. all hidden-layer representations. This allows us to derive a bound that not only considers the weights in the first linear layer but also all other linear weights. We observe in Appendix G that while MLS could be negatively correlated with the right-hand side of Equation (11), NMLS has a positive correlation with right-hand side of Equation (12) consistently (our contribution 3).

### 3.3 Local dimensionality is tied to, but not bounded by, sharpness

Now we introduce a local measure of dimensionality. Consider an input data point $\bar{\mathbf{x}}$ drawn from the training set: $\bar{\mathbf{x}} = \mathbf{x}_i$ for a specific $i \in \{1, \cdots, n\}$. Let the set of all possible perturbations around $\bar{\mathbf{x}}$ in the input space be the ball $\mathcal{B}(\bar{\mathbf{x}})_\alpha \sim \mathcal{N}(\bar{\mathbf{x}}, \alpha\mathcal{I})$, where $\alpha$ depends on the perturbation's covariance, which is given as $C_{\mathcal{B}(\bar{\mathbf{x}})} = \alpha\mathcal{I}$, with $\mathcal{I}$ as the identity matrix. We'll explore the network's representation of inputs by measuring the expansion or contraction of different aspects of the ball $\mathcal{B}(\bar{\mathbf{x}})_\alpha$ as it propagates through the network. We first propagate the ball through the network transforming each point $\mathbf{x}$ into its corresponding image $f(\mathbf{x})$. Following a Taylor expansion for points within $\mathcal{B}(\bar{\mathbf{x}})_\alpha$ as $\alpha \to 0$ with high probability we have:

$$f(\mathbf{x}) = f(\bar{\mathbf{x}}) + \nabla_{\mathbf{x}} f(\bar{\mathbf{x}}, \boldsymbol{\theta}^*)^T (\mathbf{x} - \bar{\mathbf{x}}) + O(\|\mathbf{x} - \bar{\mathbf{x}}\|^2). \tag{13}$$

We can express the limit of the covariance matrix $C_{f(\mathcal{B}(\mathbf{x}))}$ of the output $f(\mathbf{x})$ as

$$C_f^{\mathrm{lim}} := \lim_{\alpha \to 0} \frac{C_{f(\mathcal{B}(\mathbf{x})_\alpha)}}{\alpha} = \nabla_{\mathbf{x}} f(\bar{\mathbf{x}}, \boldsymbol{\theta}^*) \nabla_{\mathbf{x}}^T f(\bar{\mathbf{x}}, \boldsymbol{\theta}^*). \tag{14}$$

Our covariance expressions capture the distribution of points in $\mathcal{B}(\bar{\mathbf{x}})_\alpha$ as they go through the network $f(\bar{\mathbf{x}}, \boldsymbol{\theta}^*)$. The local Participation Ratio based on this covariance is given by:

$$D_{\mathrm{PR}}(f(\bar{\mathbf{x}})) = \lim_{\alpha \to 0} \frac{\mathrm{Tr}[C_{f(\mathcal{B}(\mathbf{x}))}]^2}{\mathrm{Tr}[(C_{f(\mathcal{B}(\mathbf{x}))})^2]} = \frac{\mathrm{Tr}[C_f^{\mathrm{lim}}]^2}{\mathrm{Tr}[(C_f^{\mathrm{lim}})^2]} \tag{15}$$

((Recanatesi et al., 2022), cf. nonlocal measures in (Gao et al., 2017; Litwin-Kumar et al., 2017; Mazzucato et al., 2016)). This quantity can be averaged across a set of samples: $D_{\mathrm{PR}}(\boldsymbol{\theta}^*) = \frac{1}{n} \sum_{i=1}^{n} D_{\mathrm{PR}}(f(\mathbf{x}_i))$. This quantity in some sense represents the sparseness of the eigenvalues of $C_f^{\mathrm{lim}}$: if we let $\boldsymbol{\lambda}$ be all the eigenvalues of $C_f^{\mathrm{lim}}$, then the local dimensionality can be written as $D_{\mathrm{PR}} = (\|\boldsymbol{\lambda}\|_1 / \|\boldsymbol{\lambda}\|_2)^2$, which attains its maximum value when all eigenvalues are equal to each other, and its minimum when all eigenvalues except for the leading one are zero. Note that the quantity retains the same value when $\boldsymbol{\lambda}$ is arbitrarily scaled. As a consequence, it is hard to find a relationship between local dimensionality and the fundamental quantity on which our bounds are based: $\|\nabla_{\mathbf{x}} f(\mathbf{x}, \boldsymbol{\theta}^*)\|_F^2$, which is $\|\boldsymbol{\lambda}\|_1$.

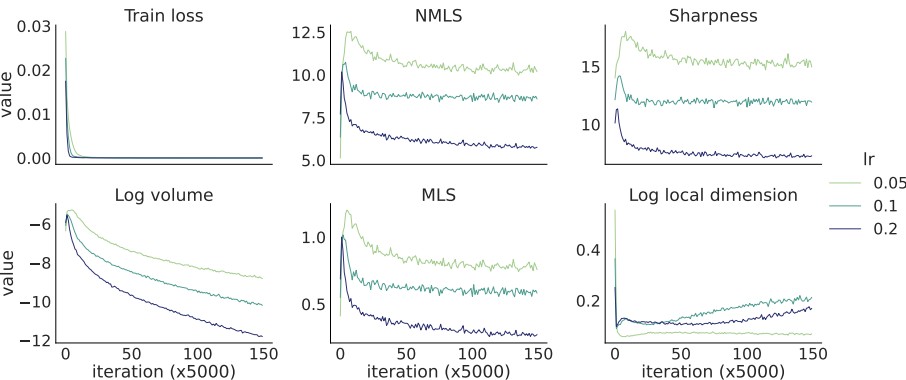

Figure 1: Trends in key variables across SGD training of the VGG-11 network with fixed batch size (equal to 20) and varying learning rates (0.05, 0.1 and 0.2). After the loss is minimized (so that an approximate interpolation solution is found) sharpness and volumes decrease together. Moreover, higher learning rates lead to lower sharpness and hence stronger compression. From left to right: training loss, NMLS, sharpness (square root of Equation (3)), log volumetric ratio (Equation (7)), MLS, and local dimensionality of the network output (Equation (15)).

### 3.4 RELATION TO REPARAMETRIZATION-INVARIANT SHARPNESS

Dinh et al. (2017) argues that a robust sharpness metric should have the reparametrization-invariant property, meaning that scaling the neighboring linear layer weights should not change the metric. While the bounds Equation (11) and Equation (12) are not strictly reparametrization-invariant, those metrics that re-design sharpness (Tsuzuku et al., 2019) to achieve invariance related to an effort to tighten our bounds (see Appendix E.1). We also evaluate the relative flatness (Petzka et al., 2021) which is also reparametrization-invariant in Appendix G.2. Another more aggressive reparametrization-invariant sharpness is proposed in Andriushchenko et al. (2023); Kwon et al. (2021), and we again show that it upper bounds input-invariant MLS in Appendix E.2. Therefore, we provide a novel perspective: reparametrization-invariant sharpness is characterized by the robustness of outputs to internal network representations.

## 4 EXPERIMENTS

### 4.1 SHARPNESS AND COMPRESSION: VERIFYING THE THEORY

The theoretical results derived above show that, during the later phase of training – the interpolation phase – measures of compression of the network's representation are upper bounded by a function of the sharpness of the loss function in parameter space. This links sharpness and representation compression: the flatter the loss landscape, the lower the upper bound on the representation's compression metrics.

However, it remains to be tested in practice whether these bounds are sufficiently tight to show a clear relationship between sharpness and representation collapse. For one such test, we conducted the following experiment. We trained a VGG-11 network (Simonyan & Zisserman, 2015) to classify images from the CIFAR-10 dataset (Krizhevsky, 2009) and calculated the sharpness (Equation (2)), the log volumetric ratio (Equation (7)), and the MLS and NMLS (Equation (11) and Equation (12)) during the training phase (Fig 1 and 2). We trained the network using SGD on images from 2 classes (out of 10) so that convergence to the interpolation regime, i.e. zero error, was faster. We explored the influence of two specific parameters that have a substantial effect on the network's training: learning rate and batch size. For each pair of learning rate and batch size parameters, we computed all quantities at hand across 100 input samples and five different random initializations for network weights.

In the first set of experiments, we began by studying the link between a decrease in sharpness during the latter phases of training and volume compression (Figure 1). We noticed that when the network reaches the interpolation regime, and the sharpness decreases, so does the volume. Similarly, the MLS decreases. All these results were consistent across multiple learning rates for a fixed batch size (of 20): specifically, for learning rates that yielded lower values of sharpness, volume was lower as well.

We then repeated the experiments while keeping the learning rate fixed (lr=0.1) and varying the batch size. The same broadly consistent trends emerged, linking a decrease in the sharpness to a compression in the representation volume (Figure 2). However, we also found that while sharpness stops decreasing after about $50 \cdot 10^3$ iterations for a batch size of 32, the volume continues to decrease as learning proceeds. This suggests that other mechanisms, beyond sharpness, may be at play in driving the compression of volumes.

We repeat the experiments with an MLP trained on the FashionMNIST dataset (Xiao et al., 2017) (Figure H.9 and Figure H.8). Although the sharpness does not noticeably decrease towards the end of the training, it follows the same trend as MLS, consistent with our bound. The volume continues to decrease after the sharpness plateaus, albeit at a much slower rate, again matching our theory while suggesting that an additional factor may be involved in its decrease.

This characterization of sharpness using compression provides a possible explanation for why (reparametrization-invariant) sharpness sometimes fails to account for the generalization behavior of the network (Wen et al., 2023; Andriushchenko et al., 2023): Compression of network output is not always desired for generalization. For example, it is observed in our Figure 1, Cohen et al. (2020) and Wu et al. (2022) that learning rate is negatively correlated with sharpness, but Wortsman et al. (2022) shows that a large learning rate can severely hurt OOD generalization performance. More intuitively, consider a scenario where one provides a large language model (LLM) with a long text sequence and instructs it to find a specific piece of information, often called the "needle in a haystack" test. Even a slight alteration in the instructions (a tiny portion of the input) given to the model should lead to a notable difference in its output depending on the desired information. Therefore, compression of network output is not a desirable property in this case.

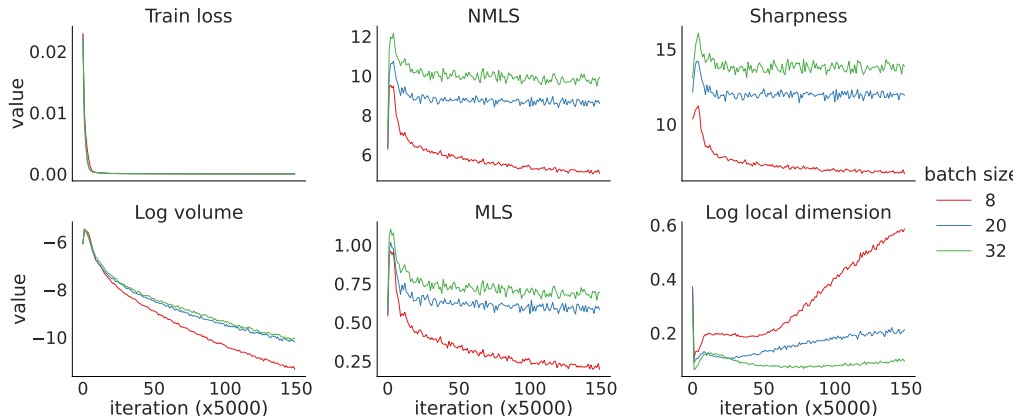

Figure 2: Trends in key variables across SGD training of the VGG-11 network with fixed learning rate size (equal to 0.1) and varying batch size (8, 20, and 32). After the loss is minimized (so that an interpolation solution is found) sharpness and volumes decrease together. Moreover, lower batch sizes lead to lower sharpness and hence stronger compression. From left to right in row-wise order: train loss, NMLS, sharpness (square root of Equation (3)), log volumetric ratio (Equation (7)), MLS, and local dimensionality of the network output (Equation (15)).

## 4.2 SHARPNESS AND LOCAL DIMENSIONALITY

Lastly, we analyze the representation's local dimensionality in a manner analogous to the analysis of volume and MLS. A priori, it is ambiguous whether the dimensionality of the data representation

should increase or decrease as the volume is compressed. For instance, the volume could decrease while maintaining its overall form and symmetry, thus preserving its dimensionality. Alternatively, one or more of the directions in the relevant tangent space could be selectively compressed, leading to an overall reduction in dimensionality.

Figures 1 and 2 show our experiments computing the local dimensionality over the course of learning. Here, we find that the local dimensionality of the representation decreases as the loss decreases to near 0, which is consistent with the viewpoint that the network compresses representations in feature space as much as possible, retaining only the directions that code for task-relevant features (Berner et al., 2020; Cohen et al., 2020). However, the local dimensionality exhibits unpredictable behavior that cannot be explained by the sharpness once the network is near the zero-loss manifold and training continues. Further experiments also demonstrate a weaker correlation of sharpness and local dimensionality compared to other metrics such as MLS and volume (Appendix Figure G.5-G.7). This discrepancy is consistent with the bounds established by our theory, which only bound the numerator of Equation (15). It is also consistent with the property of local dimensionality that we described in Section 3.3 overall: it encodes the sparseness of the eigenvalues but it does not encode the magnitude of them. This shows how local dimensionality is a distinct quality of network representations compared with volume, and is driven by mechanisms that differ from sharpness alone. We emphasize that the dimensionality we study here is a local measure, on the finest scale around a point on the "global" manifold of unit activities; dimension on larger scales (i.e., across categories or large sets of task inputs (Farrell et al., 2022; Gao et al., 2017)) may show different trends.

### 4.3 CORRELATION BETWEEN SHARPNESS AND COMPRESSION

Besides training dynamics, we also test the correlation between both sides of the bounds that we derive (see Appendix G.2). We find that MLS and the bound over NMLS, introduced in Equation (11) and Equation (12), correlate positively with NMLS in all of our experiments. Although the bound in Proposition 3.4 is loose, log volume correlates well with sharpness and MLS. Moreover, we find that sharpness is positively correlated with the generalization gap, suggesting that little reparametrization effect (Dinh et al., 2017) occurs during training; this indicates that the network weights do not change significantly, aligning with observations from Ma & Ying (2021). Finally, we found that quantities that only consider a single layer of weights such as relative flatness (Petzka et al., 2021) and the bound over MLS (Equation (11)) can exhibit a weak or even negative correlation with the generalization gap and sharpness in some cases (Figure G.6).

### 4.4 EMPIRICAL EVIDENCE IN VISION TRANSFORMERS (VITS)

Since our theory applies to linear and convolutional layers as well as residual layers (Appendix A), relationships among sharpness and compression, as demonstrated above for VGG-11 and MLP networks, should hold more generally in modern architectures such as the Vision Transformer (ViT) and its variants. While a complete characterization is beyond the scope of the current paper, we close with a first forward-looking exploration of this question. Specifically, in Figure 3 we plot the MLS normalized by the norm of the input $\frac{1}{n}\sum_{i=1}^{n} \|\mathbf{x}_i\|_2^2 \|\nabla_{\mathbf{x}_i} f\|_2^2$ against the elementwise-adaptive sharpness defined in Andriushchenko et al. (2023); Kwon et al. (2021). For large models, analytic calculation of both sharpness and MLS/NMLS that we used in previous sections is computationally infeasible, and details of the numerical approximation we used are given in Appendix F. For all the models, we attach a sigmoid layer to the output logits and use MSE loss to calculate the adaptive sharpness. Figure 3 shows the results for 181 pretrained ViT models provided by the timm package (Wightman, 2019). We observe that there is a general trend that lower sharpness indeed implies lower MLS. However, there are also outlier clusters that with data corresponding to the same model class; an interesting direction future work would be to understand the mechanisms driving this outlier behavior.

## 5 CONCLUSION

This work presents a dual perspective, uniting views in both parameter and in feature space, of several key properties of trained neural networks that have been linked to their ability to generalize. We identify two representation space quantities that are bounded by sharpness – volume compression

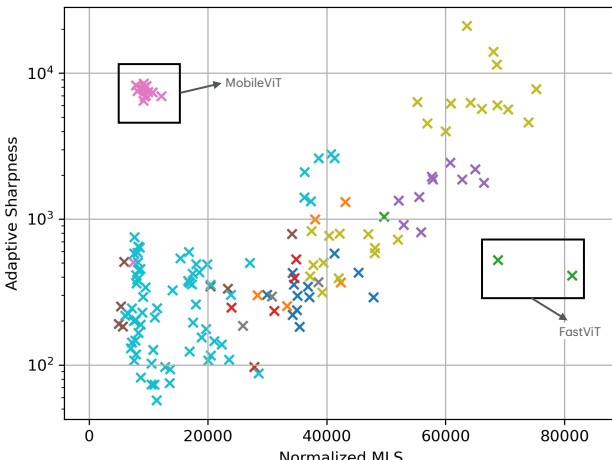

Figure 3: Adaptive sharpness vs Normalized MLS for 181 ViT models and variants. Different colors represent different model classes. For most models, there is a positive correlation between Sharpness and MLS. However, outlier clusters also exist, for MobileViT (Mehta & Rastegari, 2022) models in the upper left corner, and two FastViT (Vasu et al., 2023) models in the lower right corner.

and maximum local sensitivity – and give new explicit formulas for these bounds. We conduct experiments with both VGG-11 and MLP networks and find that the predictions of these bounds are born out for these networks, illustrating how MLS in particular is strongly correlated with sharpness. We also establish that sharpness, volume compression, and MLS are correlated, if more weakly, with test loss and hence generalization. Overall, we establish explicit links between sharpness properties in parameter spaces and compression and robustness properties in representation space.

By demonstrating both how these links can be tight, and how and when they may also become loose, we show that taking this dual perspective can bring more clarity to the often confusing question of what quantifies how well a network will generalize in practice. Indeed, many works, as reviewed in the introduction, have demonstrated how sharpness in parameter space can lead to generalization, but recent studies have established contradictory results.

This said, we view our study as a starting point to open doors between two often-distinct perspectives on generalization in neural networks. Additional theoretical and experimental research is warranted to systematically investigate the implications of our findings, with a key area being further learning problems, such as predictive learning, beyond the classification tasks studied here. Nevertheless, we are confident that highly interesting and clarifying findings lie ahead at the interface between the parameter and representation space quantities explored here.

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

## A    ADAPTATION OF INEQUALITY 6 TO RESIDUAL LAYERS

We need to slightly adapt the proof in Eq. 4 and 5. Consider a network whose first layer has a residual connection: $y = g(x + f(Wx))$, where $f$ is the nonlinearity, and $g$ is the rest of the mappings in the network. Then we have

$$\|\nabla_W g(x + f(Wx))\|_F = \|JK\|_F \|x\|_2$$
$$\nabla_x g(x + f(Wx)) = J + JKW \tag{16}$$

where $J = \frac{\partial g(x+f(Wx))}{\partial(x+f(Wx))}$ and $K = \frac{\partial f(Wx)}{\partial(Wx)}$.

Therefore, $\|\nabla_x g(x + f(Wx))\|_2 \leq \|J\|_2 + \|JK\|_2 \|W\|_2 \leq \|J\|_2 + \frac{\|\nabla_W g(x+f(Wx))\|_F}{\|x\|_2} \|W\|_2$.
Now, we get the bound for the *difference* between MLS of input and the MLS of input to the next layer:

$$\|\nabla_x g(x + f(Wx))\|_2 - \|J\|_2 \leq \frac{\|\nabla_W g(x + f(Wx))\|_F}{\|x\|_2} \|W\|_2 \tag{17}$$

Notice that if we apply this inequality to every residual layer in the network, and sum the left-hand side, we will get a telescoping sum on the left-hand side. Assuming the last layer is linear with weights $W_L$, we get $\|\nabla_x g(x + f(W_1 x))\|_2 - \|W_L\|_2 \leq \sum_{l=1}^{L-1} \frac{\|W_l\|_2}{\|x_l\|_2} \|\nabla_W g_l(x_l + f(W_l x_l))\|_F$. The right-hand side is bounded by sharpness due to Cauchy, see also Equation (33).

## B    PROOF OF EQUATION (3)

**Lemma B.1.** *If $\theta$ is an approximate interpolation solution, i.e. $\|f(\mathbf{x}_i, \theta) - \mathbf{y}_i\| < \varepsilon$ for $i \in \{1, 2, \cdots, n\}$, and second derivatives of the network function $\|\nabla^2_{\theta_j} f(\mathbf{x}_i, \theta)\| < M$ is bounded, then*

$$S(\theta^*) = \frac{1}{n} \sum_{i=1}^n \|\nabla_\theta f(\mathbf{x}_i, \theta^*)\|_F^2 + O(\varepsilon) \tag{18}$$

*Proof.* Using basic calculus we get

$$S(\theta) = \text{Tr}(\nabla^2 L(\theta))$$

$$= \frac{1}{2n} \sum_{i=1}^n \text{Tr}(\nabla^2_\theta \|f(\mathbf{x}_i, \theta) - \mathbf{y}_i\|^2)$$

$$= \frac{1}{2n} \sum_{i=1}^n \text{Tr} \, \nabla_\theta (2(f(\mathbf{x}_i, \theta) - \mathbf{y}_i)^T \nabla_\theta f(\mathbf{x}_i, \theta))$$

$$= \frac{1}{n} \sum_{i=1}^n \sum_{j=1}^m \frac{\partial}{\partial \theta_j} ((f(\mathbf{x}_i, \theta) - \mathbf{y}_i)^T \nabla_\theta f(\mathbf{x}_i, \theta))_j$$

$$= \frac{1}{n} \sum_{i=1}^n \sum_{j=1}^m \frac{\partial}{\partial \theta_j} (f(\mathbf{x}_i, \theta) - \mathbf{y}_i)^T \nabla_{\theta_j} f(\mathbf{x}_i, \theta)$$

$$= \frac{1}{n} \sum_{i=1}^n \sum_{j=1}^m \|\nabla_{\theta_j} f(\mathbf{x}_i, \theta)\|_2^2 + (f(\mathbf{x}_i, \theta) - \mathbf{y}_i)^T \nabla^2_{\theta_j} f(\mathbf{x}_i, \theta)$$

$$= \frac{1}{n} \sum_{i=1}^n \|\nabla_\theta f(\mathbf{x}_i, \theta)\|_F^2 + \frac{1}{n} \sum_{i=1}^n \sum_{j=1}^m (f(\mathbf{x}_i, \theta) - \mathbf{y}_i)^T \nabla^2_{\theta_j} f(\mathbf{x}_i, \theta).$$

Therefore

$$\left| S(\theta) - \frac{1}{n} \sum_{i=1}^n \|\nabla_\theta f(\mathbf{x}_i, \theta)\|_F^2 \right| < \frac{1}{n} \sum_{i=1}^n \sum_{j=1}^m |(f(\mathbf{x}_i, \theta) - \mathbf{y}_i)^T \nabla^2_{\theta_j} f(\mathbf{x}_i, \theta)| < mM\varepsilon = O(\varepsilon).$$

$$\tag{19}$$

$\square$

In other words, when the network reaches zero training error and enters the interpolation phase (i.e. it classifies all training data correctly), Equation (3) will be a good enough approximation of the sharpness because the quadratic training loss is sufficiently small.

## C    PROOF OF PROPOSITION 3.4 AND PROPOSITION 3.6

For notation simplicity, we write $f_i := f(\mathbf{x}_i, \boldsymbol{\theta}^*)$ in what follows. Because of Equation (5), we have the following inequality due to Cauchy-Swartz inequality,

$$\frac{1}{n}\sum_{i=1}^{n}\|\nabla_{\mathbf{x}}f_i\|_F^k \le \|\mathbf{W}\|_2^k \frac{1}{n}\sum_{i=1}^{n}\frac{\|\nabla_{\mathbf{W}}f_i\|_F^k}{\|\mathbf{x}_i\|_2^k}$$

$$\le \frac{1}{n}\|\mathbf{W}\|_2^k\sqrt{\sum_{i=1}^{n}\frac{1}{\|\mathbf{x}_i\|_2^{2k}}} \cdot \sqrt{\sum_{i=1}^{n}\|\nabla_{\mathbf{W}}f_i\|_F^{2k}}. \tag{20}$$

Since the input weights $\mathbf{W}$ is just a part of all the weights ($\boldsymbol{\theta}$) of the network, we have $\|\nabla_{\mathbf{W}}f_i\|_F^k \le \|\nabla_{\boldsymbol{\theta}}f_i\|_F^k$.

We next show the correctness of Proposition 3.4 with a standard lemma.

**Lemma C.1.** *For vector* $\mathbf{x}$, $\|\mathbf{x}\|_p \ge \|\mathbf{x}\|_q$ *for* $1 \le p \le q \le \infty$.

*Proof.* First we show that for $0 < k < 1$, we have $(|a| + |b|)^k \le |a|^k + |b|^k$. It's trivial when either $a$ or $b$ is 0. So W.L.O.G, we can assume that $|a| < |b|$, and divide both sides by $|b|^k$. Therefore it suffices to show that for $0 < t < 1$, $(1 + t)^k < t^k + 1$. Let $f(t) = (1 + t)^k - t^k - 1$, then $f(0) = 0$, and $f'(t) = k(1 + t)^{k-1} - kt^{k-1}$. Because $k - 1 < 0$, $1 + t > 1$ and $t < 1$, $t^{k-1} > 1 > (1 + t)^{k-1}$. Therefore $f'(t) < 0$ and $f(t) < 0$ for $0 < t < 1$. Combining all cases, we have $(|a| + |b|)^k \le |a|^k + |b|^k$ for $0 < k < 1$. By induction, we have $(\sum_n |a_n|)^k \le \sum_n |a_n|^k$.

Now we can prove the lemma using the conclusion above,

$$\left(\sum_n |x_n|^q\right)^{1/q} = \left(\sum_n |x_n|^q\right)^{p/q \cdot 1/p} \le \left(\sum_n (|x_n|^q)^{p/q}\right)^{1/p} = \left(\sum_n |x_n|^p\right)^{1/p}$$

$\square$

Now we can prove Proposition 3.4

**Proposition.** *The local volumetric ratio is upper bounded by a sharpness related quantity:*

$$dV_{f(\boldsymbol{\theta}^*)} \le \frac{N^{-N/2}}{n}\sum_{i=1}^{n}\|\nabla_{\mathbf{x}}f(\mathbf{x}, \boldsymbol{\theta}^*)\|_F^N \le \frac{1}{n}\sqrt{\sum_{i=1}^{n}\frac{\|\mathbf{W}\|_2^{2N}}{\|\mathbf{x}_i\|_2^{2N}}\left(\frac{nS(\boldsymbol{\theta}^*)}{N}\right)^{N/2}} \tag{21}$$

*for all* $N \ge 1$.

*Proof.* Take the $x_i$ in Lemma C.1 to be $\|\nabla_{\boldsymbol{\theta}}f(\mathbf{x}_i, \boldsymbol{\theta}^*)\|_F^2$ and let $p = 1, q = k$, then we get

$$\left(\sum_{i=1}^{n}(\|\nabla_{\boldsymbol{\theta}}f_i\|_F^2)^k\right)^{1/k} \le \sum_{i=1}^{n}\|\nabla_{\boldsymbol{\theta}}f_i\|_F^2. \tag{22}$$

Therefore,

$$\frac{1}{n}\|\mathbf{W}\|_2^k\sqrt{\sum_{i=1}^{n}\frac{1}{\|\mathbf{x}_i\|_2^{2k}}} \cdot \sqrt{\sum_{i=1}^{n}\|\nabla_{\mathbf{W}}f_i\|_F^{2k}} \le n^{k/2-1}\|\mathbf{W}\|_2^k\sqrt{\sum_{i=1}^{n}\frac{1}{\|\mathbf{x}_i\|_2^{2k}}}\left(\frac{1}{n}\sum_{i=1}^{n}\|\nabla_{\boldsymbol{\theta}}f_i\|_F^2\right)^{k/2}$$

$$= n^{k/2-1}\|\mathbf{W}\|_2^k\sqrt{\sum_{i=1}^{n}\frac{1}{\|\mathbf{x}_i\|_2^{2k}}}S(\boldsymbol{\theta}^*)^{k/2}$$

$$\tag{23}$$

$\square$

Next, we show that the first inequality in Equation (23) can be tightened by considering all linear layer weights.

**Proposition.** *The network volumetric ratio is upper bounded by a sharpness related quantity:*

$$\sum_{l=1}^{L} dV_{f_l} \leq \frac{N^{-N/2}}{n} \sum_{l=1}^{L} \sum_{i=1}^{n} \|\nabla_{\mathbf{x}^l} f_i^l\|_F^N \leq \frac{1}{n} \sqrt{\sum_{l=1}^{L} \sum_{i=1}^{n} \frac{\|\mathbf{W}_l\|_2^{2N}}{\|\mathbf{x}_i^l\|_2^{2N}} \cdot \left(\frac{nS(\boldsymbol{\theta}^*)}{N}\right)^{N/2}}. \quad (24)$$

*Proof.* Recall that the input to $l$-th linear layer as $x_i^l$ for $l = 1, 2, \cdots, L$. In particular, $x_i^1$ is the input of the entire network. Similarly, $\mathbf{W}_l$ is the weight matrix of $l$-th linear/convolutional layer. With a slight abuse of notation, we use $f^l$ to denote the mapping from the activity of $l$-th layer to the final output, and $f_i^l := f^l(\mathbf{x}_i, \boldsymbol{\theta}^*)$. We can apply Cauchy-Swartz inequality again to get

$$
\begin{aligned}
\frac{1}{n} \sum_{l=1}^{L} \sum_{i=1}^{n} \|\nabla_{\mathbf{x}^l} f_i^l\|_F^k &\leq \frac{1}{n} \sum_{l=1}^{L} \sqrt{\sum_{i=1}^{n} \frac{\|\mathbf{W}_l\|_2^{2k}}{\|\mathbf{x}_i^l\|_2^{2k}}} \cdot \sqrt{\sum_{i=1}^{n} \|\nabla_{\mathbf{W}_l} f_i^l\|_F^{2k}} \\
&\leq \sqrt{\frac{1}{n} \sum_{l=1}^{L} \sum_{i=1}^{n} \frac{\|\mathbf{W}_l\|_2^{2k}}{\|\mathbf{x}_i^l\|_2^{2k}}} \cdot \sqrt{\frac{1}{n} \sum_{l=1}^{L} \sum_{i=1}^{n} \|\nabla_{\mathbf{W}_l} f_i^l\|_F^{2k}}.
\end{aligned}
\quad (25)
$$

Using Lemma C.1 again we have

$$
\begin{aligned}
\left(\sum_{l=1}^{L} (\|\nabla_{\boldsymbol{W}_l} f_i^l\|_F^2)^k\right)^{1/k} &\leq \sum_{l=1}^{L} \|\nabla_{\boldsymbol{W}_l} f_i^l\|_F^2 = \|\nabla_{\boldsymbol{\theta}} f_i\|_F^2, \\
\left(\sum_{i=1}^{n} (\|\nabla_{\boldsymbol{\theta}} f_i\|_F^2)^k\right)^{1/k} &\leq \sum_{i=1}^{n} \|\nabla_{\boldsymbol{\theta}} f_i\|_F^2 = nS(\boldsymbol{\theta}^*),
\end{aligned}
\quad (26)
$$

The second equality holds because both sides represent the same gradients in the computation graph. Therefore from Equation (25), we have

$$\frac{1}{n} \sum_{l=1}^{L} \sum_{i=1}^{n} \|\nabla_{\mathbf{x}^l} f_i^l\|_F^k \leq \sqrt{\frac{1}{n} \sum_{l=1}^{L} \sum_{i=1}^{n} \frac{\|\mathbf{W}_l\|_2^{2k}}{\|\mathbf{x}_i^l\|_2^{2k}}} \cdot \sqrt{n^{k-1} S(\boldsymbol{\theta}^*)^k} \quad (27)$$

$\square$

# D PROOF OF PROPOSITION 3.8 AND PROPOSITION 3.10

Below we give the proof of Proposition 3.8.

**Proposition.** *The maximum local sensitivity is upper bounded by a sharpness related quantity:*

$$\text{MLS} = \frac{1}{n} \sum_{i=1}^{n} \|\nabla_{\mathbf{x}} f(\mathbf{x}_i, \boldsymbol{\theta}^*)\|_2 \leq \|\mathbf{W}\|_2 \sqrt{\frac{1}{n} \sum_{i=1}^{n} \frac{1}{\|\mathbf{x}_i\|_2^2}} S(\boldsymbol{\theta}^*)^{1/2}. \quad (28)$$

*Proof.* From Equation (5), we get

$$\text{MLS} = \frac{1}{n} \sum_{i=1}^{n} \|\nabla_{\mathbf{x}} f_i\|_2 \leq \|\mathbf{W}\|_2 \frac{1}{n} \sum_{i=1}^{n} \frac{\|\nabla_{\mathbf{w}} f_i\|_F}{\|\mathbf{x}_i\|_2}. \quad (29)$$

Now the Cauchy-Schwarz inequality tells us that

$$\left(\sum_{i=1}^{n} \frac{\|\nabla_{\mathbf{w}} f_i\|}{\|\mathbf{x}_i\|_2}\right)^2 \leq \left(\sum_{i=1}^{n} \frac{1}{\|\mathbf{x}_i\|_2^2}\right) \cdot \left(\sum_{i=1}^{n} \|\nabla_{\mathbf{w}} f_i\|_F^2\right). \quad (30)$$

Therefore

$$
\begin{aligned}
\mathrm{MLS} &\leq \|\mathbf{W}\|_2 \sqrt{\frac{1}{n}\sum_{i=1}^{n}\frac{1}{\|\mathbf{x}_i\|_2^2}} \cdot \sqrt{\frac{1}{n}\sum_{i=1}^{n}\|\nabla_{\mathbf{W}} f_i\|_F^2} \\
&\leq \|\mathbf{W}\|_2 \sqrt{\frac{1}{n}\sum_{i=1}^{n}\frac{1}{\|\mathbf{x}_i\|_2^2}} \cdot S(\boldsymbol{\theta}^*)^{1/2}.
\end{aligned}
\tag{31}
$$

$\square$

Now we can prove Proposition 3.10.

**Proposition.** *The network maximum local sensitivity is upper bounded by a sharpness related quantity:*

$$
\mathrm{NMLS} = \frac{1}{n}\sum_{l=1}^{L}\sum_{i=1}^{n}\|\nabla_{\mathbf{x}^l} f^l(\mathbf{x}_i^l, \boldsymbol{\theta}^*)\|_2 \leq \sqrt{\frac{1}{n}\sum_{i=1}^{n}\sum_{l=1}^{L}\frac{\|\mathbf{W}_l\|_2^2}{\|\mathbf{x}_i^l\|^2}} \cdot S(\boldsymbol{\theta}^*)^{1/2}.
\tag{32}
$$

*Proof.* We can apply Equation (31) to every linear layer and again apply the Cauchy-Schwarz inequality to obtain

$$
\begin{aligned}
\mathrm{NMLS} &= \frac{1}{n}\sum_{l=1}^{L}\sum_{i=1}^{n}\|\nabla_{\mathbf{x}} f_l(\mathbf{x}_i^l, \boldsymbol{\theta}^*)\|_2 \\
&\leq \sum_{l=1}^{L}\left(\sqrt{\frac{1}{n}\sum_{i=1}^{n}\frac{\|\mathbf{W}_l\|_2^2}{\|\mathbf{x}_i^l\|_2^2}}\sqrt{\frac{1}{n}\sum_{i=1}^{n}\|\nabla_{\mathbf{W}_l} f_i^l\|_F^2}\right) \\
&\leq \sqrt{\frac{1}{n}\sum_{i=1}^{n}\sum_{l=1}^{L}\frac{\|\mathbf{W}_l\|_2^2}{\|\mathbf{x}_i^l\|_2^2}}\sqrt{\frac{1}{n}\sum_{i=1}^{n}\sum_{l=1}^{L}\|\nabla_{\mathbf{W}_l} f_i^l\|_F^2} \\
&\leq \sqrt{\frac{1}{n}\sum_{i=1}^{n}\sum_{l=1}^{L}\frac{\|\mathbf{W}_l\|_2^2}{\|\mathbf{x}_i^l\|_2^2}} \cdot S(\boldsymbol{\theta}^*)^{1/2}.
\end{aligned}
\tag{33}
$$

Note that the gap in the last inequality is significantly smaller than that of Equation (31) since now we consider all linear weights. $\square$

# E    REPARAMETRIZATION-INVARIANT SHARPNESS AND INPUT-INVARIANT MLS

## E.1    REPARAMETRIZATION-INVARIANT SHARPNESS IN TSUZUKU ET AL. (2019)

In this appendix, we show that the reparametrization-invariant sharpness metrics introduced in Tsuzuku et al. (2019) can seen as an effort to tighten the bound that we derived above. For matrix-normalized sharpness (cf. Equation 13), the connection is immediately seen from Equation (31). Let

$$
\overline{\mathbf{x}} = \left(\frac{1}{n}\sum_{i=1}^{n}\frac{1}{\|\mathbf{x}_i\|_2^2}\right)^{-\frac{1}{2}}.
\tag{34}
$$

Then from Equation (31) we have

$$
\sum_{l=1}^{L}\overline{\mathbf{x}}^l \cdot \mathrm{MLS}^l \leq \sum_{l=1}^{L}\|\mathbf{W}_l\|_2\sqrt{\frac{1}{n}\sum_{i=1}^{n}\|\nabla_{\mathbf{W}_l} f_i^l\|_F^2} \approx \sum_{l=1}^{L}\|\mathbf{W}_l\|_2\sqrt{S(\mathbf{W}_l)},
\tag{35}
$$

where $S(\mathbf{W}_l)$ is the trace of Hessian of the loss w.r.t. the weights of the $l$-th layer. The right-hand side of Equation (35) is exactly what Tsuzuku et al. (2019) refer to as the matrix-normalized sharpness. Note that a similar inequality holds if we use Frobenius norm instead of 2-norm of the weights.

Tsuzuku et al. (2019) also pose an interesting optimization problem (cf. Equation 17) to define the normalized sharpness:

$$\min_{\boldsymbol{\sigma},\boldsymbol{\sigma}'} \sum_{i,j} \left( \frac{\partial^2 L}{\partial W_{i,j} \partial W_{i,j}} (\sigma_i \sigma_j')^2 + \frac{W_{i,j}^2}{4\lambda^2 (\sigma_i \sigma_j')^2} \right). \tag{36}$$

Note that by Lemma B.1, $\frac{\partial^2 L}{\partial W_{i,j} \partial W_{i,j}} \approx \|\nabla_{\mathbf{W}_{i,j}} f\|_2^2$. Moreover, we have

$$\sum_{i,j} \left( \|\nabla_{\mathbf{W}_{i,j}} f\|^2 (\sigma_i \sigma_j')^2 + \frac{W_{i,j}^2}{4\lambda^2 (\sigma_i \sigma_j')^2} \right) \geq \frac{1}{\lambda} \sqrt{\sum_{i,j} (\nabla_{\mathbf{W}_{i,j}} f)^2 (\sigma_i \sigma_j')^2} \cdot \sqrt{\sum_{i,j} \frac{W_{i,j}^2}{(\sigma_i \sigma_j')^2}}$$

$$\geq \frac{1}{\lambda} \|\operatorname{diag}(\boldsymbol{\sigma}) J\|_F \|\operatorname{diag}(\boldsymbol{\sigma}') \mathbf{x}\|_2 \|\operatorname{diag}(\boldsymbol{\sigma}^{-1}) \mathbf{W} \operatorname{diag}(\boldsymbol{\sigma}'^{-1})\|_F$$

$$\geq \frac{1}{\lambda} \|\operatorname{diag}(\boldsymbol{\sigma}'^{-1}) W^T J\|_F \|\operatorname{diag}(\boldsymbol{\sigma}') \mathbf{x}\|_2$$

$$= \frac{1}{\lambda} \|\operatorname{diag}(\boldsymbol{\sigma}'^{-1}) \nabla_{\mathbf{x}} f\|_F \|\operatorname{diag}(\boldsymbol{\sigma}') \mathbf{x}\|_2, \tag{37}$$

where $J = \frac{\partial f(\mathbf{W}\mathbf{x}; \bar{\boldsymbol{\theta}})}{\partial (\mathbf{W}\mathbf{x})}$ (see some of the calculations in Equation (4)). Therefore, the optimization problem Equation (36) is equivalent to choosing $\boldsymbol{\sigma}, \boldsymbol{\sigma}'$ to minimize the upper bound on a scale-invariant MLS-like quantity (the quantity is invariant under the transformation of the first layer from $\mathbf{W}\mathbf{x}$ to $\mathbf{W} \operatorname{diag}(\boldsymbol{\sigma}^{-1})(\operatorname{diag}(\boldsymbol{\sigma})\mathbf{x})$, where $\operatorname{diag}(\boldsymbol{\sigma})\mathbf{x}$ becomes the new input). For simplicity, we do not scale the original dataset in our work and only compare MLS within the same dataset. As a result, we can characterize those reparametrization-invariant sharpness metrics by the robustness of output to the input. If we consider all linear weights in the network, then those metrics indicate the robustness of output to internal network representations.

### E.2 REPARAMETRIZATION-INVARIANT SHARPNESS UPPER-BOUNDS INPUT-INVARIANT MLS

In this appendix, we consider the adaptive average-case n-sharpness considered in Kwon et al. (2021); Andriushchenko et al. (2023):

$$S_{avg}^\rho(\mathbf{w}, |\mathbf{w}|) \triangleq \frac{2}{\rho^2} \mathbb{E}_{S \sim P_n, \delta \sim \mathcal{N}(0, \rho^2 \operatorname{diag}(|\mathbf{w}|^2))} \left[ L_S(\mathbf{w} + \delta) - L_S(\mathbf{w}) \right], \tag{38}$$

which is shown to be *elementwise* adaptive sharpness in Andriushchenko et al. (2023). They also show that for a thrice differentiable loss, $L(w)$, the average-case elementwise adaptive sharpness can be written as

$$S_{avg}^\rho(\mathbf{w}, |\mathbf{w}|) = \mathbb{E}_{S \sim P_n} \left[ \operatorname{Tr} \left( \nabla^2 L_S(\mathbf{w}) \odot |\mathbf{w}||\mathbf{w}|^\top \right) \right] + O(\rho). \tag{39}$$

**Definition E.1.** *We define the **Elementwise-Adaptive Sharpness** $S_{adaptive}$ to be*

$$S_{adaptive}(\mathbf{w}) \triangleq \lim_{\rho \to 0} S_{avg}^\rho(\mathbf{w}, |\mathbf{w}|) = \mathbb{E}_{S \sim P_n} \left[ \operatorname{Tr} \left( \nabla^2 L_S(\mathbf{w}) \odot |\mathbf{w}||\mathbf{w}|^\top \right) \right] \tag{40}$$

In this appendix, we focus on the property of $S_{\text{adaptive}}$ instead of the approximation Equation (39). Adapting the proof of $Lemma\ B.1$, we have the following lemma.

**Lemma E.2.** *If $\boldsymbol{\theta}$ is an approximate interpolation solution, i.e. $\|f(\mathbf{x}_i, \boldsymbol{\theta}) - \mathbf{y}_i\| < \varepsilon$ for $i \in \{1, 2, \cdots, n\}$, and $|\boldsymbol{\theta}_j|^2 \|\nabla_{\theta_j}^2 f(\mathbf{x}_i, \boldsymbol{\theta})\| < M$ for all $j$, then*

$$S_{adaptive}(\boldsymbol{\theta}^*) = \frac{1}{n} \sum_{i=1}^n \sum_{j=1}^m |\boldsymbol{\theta}_j|^2 \left\| \nabla_{\boldsymbol{\theta}_j} f(\mathbf{x}_i, \boldsymbol{\theta}) \right\|_2^2 + O(\varepsilon), \tag{41}$$

*where $m$ is the number of parameters.*

*Proof.* Using basic calculus we get

$$S_{\text{adaptive}}(\boldsymbol{\theta}) = \frac{1}{2n} \sum_{i=1}^{n} \text{Tr}(\nabla_{\boldsymbol{\theta}}^2 \| f(\mathbf{x}_i, \boldsymbol{\theta}) - \mathbf{y}_i \|^2 \odot |\boldsymbol{\theta}||\boldsymbol{\theta}|^\top)$$

$$= \frac{1}{2n} \sum_{i=1}^{n} \text{Tr} \, \nabla_{\boldsymbol{\theta}} (2(f(\mathbf{x}_i, \boldsymbol{\theta}) - \mathbf{y}_i)^T \nabla_{\boldsymbol{\theta}} f(\mathbf{x}_i, \boldsymbol{\theta})) \odot |\boldsymbol{\theta}||\boldsymbol{\theta}|^\top$$

$$= \frac{1}{n} \sum_{i=1}^{n} \sum_{j=1}^{m} |\boldsymbol{\theta}_j|^2 \frac{\partial}{\partial \boldsymbol{\theta}_j} ((f(\mathbf{x}_i, \boldsymbol{\theta}) - \mathbf{y}_i)^T \nabla_{\boldsymbol{\theta}} f(\mathbf{x}_i, \boldsymbol{\theta}))_j$$

$$= \frac{1}{n} \sum_{i=1}^{n} \sum_{j=1}^{m} |\boldsymbol{\theta}_j|^2 \frac{\partial}{\partial \boldsymbol{\theta}_j} (f(\mathbf{x}_i, \boldsymbol{\theta}) - \mathbf{y}_i)^T \nabla_{\boldsymbol{\theta}_j} f(\mathbf{x}_i, \boldsymbol{\theta})$$

$$= \frac{1}{n} \sum_{i=1}^{n} \sum_{j=1}^{m} |\boldsymbol{\theta}_j|^2 \left\| \nabla_{\boldsymbol{\theta}_j} f(\mathbf{x}_i, \boldsymbol{\theta}) \right\|_2^2 + |\boldsymbol{\theta}_j|^2 (f(\mathbf{x}_i, \boldsymbol{\theta}) - \mathbf{y}_i)^T \nabla_{\boldsymbol{\theta}_j}^2 f(\mathbf{x}_i, \boldsymbol{\theta})$$

$$= \frac{1}{n} \sum_{i=1}^{n} \sum_{j=1}^{m} |\boldsymbol{\theta}_j|^2 \left\| \nabla_{\boldsymbol{\theta}_j} f(\mathbf{x}_i, \boldsymbol{\theta}) \right\|_2^2 + \frac{1}{n} \sum_{i=1}^{n} \sum_{j=1}^{m} |\boldsymbol{\theta}_j|^2 (f(\mathbf{x}_i, \boldsymbol{\theta}) - \mathbf{y}_i)^T \nabla_{\boldsymbol{\theta}_j}^2 f(\mathbf{x}_i, \boldsymbol{\theta})$$

Therefore

$$\left| S(\boldsymbol{\theta}) - \frac{1}{n} \sum_{i=1}^{n} \| \nabla_{\boldsymbol{\theta}} f(\mathbf{x}_i, \boldsymbol{\theta}) \|_F^2 \right| < \frac{1}{n} \sum_{i=1}^{n} \sum_{j=1}^{m} |(f(\mathbf{x}_i, \boldsymbol{\theta}) - \mathbf{y}_i)^T |\boldsymbol{\theta}_j|^2 \nabla_{\boldsymbol{\theta}_j}^2 f(\mathbf{x}_i, \boldsymbol{\theta})| < mM\varepsilon = O(\varepsilon).$$

$$(42)$$

$\square$

**Definition E.3.** *We define the **Input-invariant MLS** of a network $f : \mathbb{R}^N \to \mathbb{R}^M$ to be*

$$\frac{1}{n} \sum_{i=1}^{n} \sum_{p=1}^{N} \left\| \nabla_{x_p^{(i)}} f \right\|_2^2 (x_p^{(i)})^2 \,, \tag{43}$$

*where $x_p^{(i)}$ is the p-th entry of i-th training sample.*

It turns out that again the adaptive sharpness upper bounds the input-invariant MLS.

**Proposition E.4.** *Assuming that the condition of Lemma E.2 holds, then reparametrization-invariant sharpness upper-bounds input-invariant MLS:*

$$\frac{1}{n} \sum_{i=1}^{n} \sum_{j=1}^{m} |\boldsymbol{\theta}_j|^2 \left\| \nabla_{\boldsymbol{\theta}_j} f(\mathbf{x}_i, \boldsymbol{\theta}) \right\|_2^2 \geq \frac{1}{nd} \sum_{i=1}^{n} \sum_{p=1}^{N} \left\| \nabla_{x_p^{(i)}} f \right\|_2^2 (x_p^{(i)})^2 \tag{44}$$

*Proof.* Now we adapt the linear stability trick. For $\boldsymbol{\theta} = \mathbf{W}$ the first layer weight, we have

$$\sum_{j=1}^{m} |\boldsymbol{\theta}_j|^2 \left\| \nabla_{\boldsymbol{\theta}_j} f(\mathbf{x}, \boldsymbol{\theta}) \right\|_2^2 = \sum_{i,j,k} J_{jk}^2 \mathbf{W}_{ki}^2 x_p^2$$

$$= \sum_{i,j} \left( \sum_{k=1}^{d} J_{jk}^2 \mathbf{W}_{ki}^2 \right) x_p^2 \tag{45}$$

$$\geq \frac{1}{d} \sum_i \| \nabla_{\mathbf{x}_i} f \|_2^2 \mathbf{x}_i^2$$

where same as in Equation (4), $J = \frac{\partial f(\mathbf{W}\mathbf{x}; \bar{\boldsymbol{\theta}})}{\partial (\mathbf{W}\mathbf{x})}$, $\nabla_{\mathbf{x}} f(\mathbf{W}\mathbf{x}; \bar{\boldsymbol{\theta}}) = J\mathbf{W}$, and $x_p$ is the p-th entry of $\mathbf{x}$. Taking the sample mean of both sides proves the proposition.

$\square$

# F   NUMERICAL APPROXIMATION OF MLS AND ELEMENTWISE-ADAPTIVE SHARPNESS

In this appendix, we detail how we approximate the normalized MLS and adaptive sharpness in Section 4.4. Note that for all network $f$ the last layer is the sigmoid function, so the output is bounded in $(0, 1)$, and we use MSE loss to be consistent with the rest of the paper.

For the adaptive sharpness, we adopt the definition in Andriushchenko et al. (2023) and uses sample mean to approximate the expectation in Equation (38). Therefore, for network $f(\mathbf{w})$,

$$S_{\text{adaptive}}(f) = \frac{1}{nm} \sum_{i=1}^{n} \sum_{j=1}^{m} L(\mathbf{x}_i; \mathbf{w} + \delta_j) - L(\mathbf{x}_i; \mathbf{w}), \tag{46}$$

where $\delta \sim \mathcal{N}(0, 0.01 \operatorname{diag}(|w|^2))$.

For normalized MLS, we first reiterate the definition.

**Definition F.1.** *We define the **normalized MLS** as $\frac{1}{n} \sum_{i=1}^{n} \|\mathbf{x}_i\|_2^2 \|\nabla_{\mathbf{x}_i} f\|_2^2$*

Therefore, to approximate normalized MLS, we need to approximate $\|\nabla_{\mathbf{x}_i} f\|_2$. By definition of matrix 2-norm,

$$\|\nabla_{\mathbf{x}} f\|_2 = \sup_{\delta} \frac{\|\nabla_{\mathbf{x}} f \, \delta\|_2}{\|\delta\|_2} \approx \max_{\delta} \frac{\|f(\mathbf{x} + \delta) - f(\mathbf{x})\|_2}{\|\delta\|_2}. \tag{47}$$

To solve this optimization problem, we start from a randomly sampled vector $\delta$ that has the same shape as the network input, and we update $\delta$ using gradient descent.

# G   EMPIRICAL ANALYSIS OF THE BOUND

## G.1   TIGHTNESS OF THE BOUND

In this section, we mainly explore the tightness of the bound in Equation (11) for reasons discussed in Section 3.2. First we rewrite Equation (11) as

$$
\begin{aligned}
\text{MLS} &= \frac{1}{n} \sum_{i=1}^{n} \|\nabla_{\mathbf{x}} f(\mathbf{x}_i, \boldsymbol{\theta}^*)\|_2 &&:= A \\
&\leq \frac{\|\mathbf{W}\|_2}{n} \sum_{i=1}^{n} \frac{\|\nabla_{\mathbf{w}} f(\mathbf{x}_i, \boldsymbol{\theta}^*)\|_F}{\|\mathbf{x}_i\|_2} &&:= B \\
&\leq \|\mathbf{W}\|_2 \sqrt{\frac{1}{n} \sum_{i=1}^{n} \frac{1}{\|\mathbf{x}_i\|_2^2}} \sqrt{\frac{1}{n} \sum_{i=1}^{n} \|\nabla_{\mathbf{w}} f(\mathbf{x}_i, \boldsymbol{\theta}^*)\|_F^2} &&:= C \\
&\leq \|\mathbf{W}\|_2 \sqrt{\frac{1}{n} \sum_{i=1}^{n} \frac{1}{\|\mathbf{x}_i\|_2^2}} S(\boldsymbol{\theta}^*)^{1/2} &&:= D
\end{aligned}
\tag{48}
$$

Thus Equation (11) consists of 3 different steps of relaxations. We analyze them one by one:

1. $(A \leq B)$ The equality holds when $\|W^T J\|_2 = \|W\|_2 \|J\|_2$ and $\|J\|_F = \|J\|_2$, where $J = \frac{\partial f(\mathbf{W}\mathbf{x}; \bar{\boldsymbol{\theta}})}{\partial (\mathbf{W}\mathbf{x})}$. The former equality requires that $W$ and $J$ have the same left singular vectors. The latter requires $J$ to have zero singular values except for the largest singular value. Since $J$ depends on the specific neural network architecture and training process, we test the tightness of this bound empirically (Figure G.4).

2. $(B \leq C)$ The equality requires $\frac{\|\nabla_{\mathbf{w}} f(\mathbf{x}_i, \boldsymbol{\theta}^*)\|_F}{\|\mathbf{x}_i\|_2}$ to be the same for all $i$. In other words, the bound is tight when $\frac{\|\nabla_{\mathbf{w}} f(\mathbf{x}_i, \boldsymbol{\theta}^*)\|_F}{\|\mathbf{x}_i\|_2}$ does not vary too much from sample to sample.

3. $(C \leq D)$ The equality holds if the model is linear, i.e. $\boldsymbol{\theta} = \mathbf{W}$.

We empirically verify the tightness of the above bounds in Figure G.4

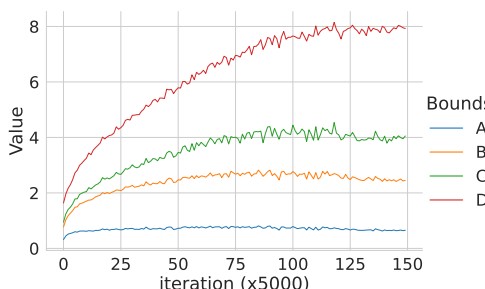

Figure G.4: **Empirical tightness of the bounds.** We empirically verify that the inequalities in Equation (48) hold and test their tightness. The results are shown for a fully connected feedforward network trained on the FashionMNIST dataset. The quantities A, B, C, and D are defined in Equation (48). We see that the gap between C and D is large compared to the gap between A and B or B and C. This indicates that partial sharpness $\|\nabla_{\mathbf{W}} f(\mathbf{x}_i, \boldsymbol{\theta}^*)\|_F$ (sensitivity of the loss w.r.t. only the input weights) is more indicative of the change in the maximum local sensitivity (A). Indeed, correlation analysis shows that bound C is positively correlated with MLS while bound D, perhaps surprisingly, is negatively correlated with MLS (Figure G.6).

## G.2 CORRELATION ANALYSIS

We empirically show how different metrics correlate with each other, and how these correlations can be predicted from our bounds. We train 100 VGG-11 networks with different batch sizes, learning rates, and random initialization to classify images from the CIFAR-10 dataset, and plot pairwise scatter plots between different quantities at the end of the training: local dimensionality, sharpness (square root of Equation (3)), log volume (Equation (7)), MLS (Equation (11)), NMLS (Equation (12)), generalization gap (gen gap), D (Equation (48)), bound (right-hand side of Equation (12)) and relative sharpness (Petzka et al., 2021) (see Figure G.5). We only include CIFAR-10 data with 2 labels to ensure that the final training accuracy is close to 100%.

We repeat the analysis on MLPs and LeNets trained on the FashionMNIST dataset and the CIFAR-10 dataset (Figure G.6 and Figure G.7). We find that

1. The bound over NMLS, MLS, and NMLS introduced in Equation (12) and Equation (11) consistently correlates positively with the generalization gap.

2. Although the bound in Equation (9) is loose, log volume correlates well with sharpness and MLS.

3. Sharpness is positively correlated with the generalization gap, indicating that little reparametrization effect (Dinh et al., 2017) is happening during training, i.e. the network weights do not change too much during training. This is consistent with observations in Ma & Ying (2021).

4. The bound derived in Equation (12) correlates positively with NMLS in all experiments.

5. MLS that only consider the first layer weights can sometimes negatively correlate with the bound derived in Equation (11) (Figure G.6).

6. Relative flatness that only consider the last layer weights introduced in (Petzka et al., 2021) shows weak (even negative) correlation with the generalization gap. Note that "relative flatness" is a misnomer that is easier understood as "relative *sharpness*", and is supposed to be *positively* correlated with the generalization gap.

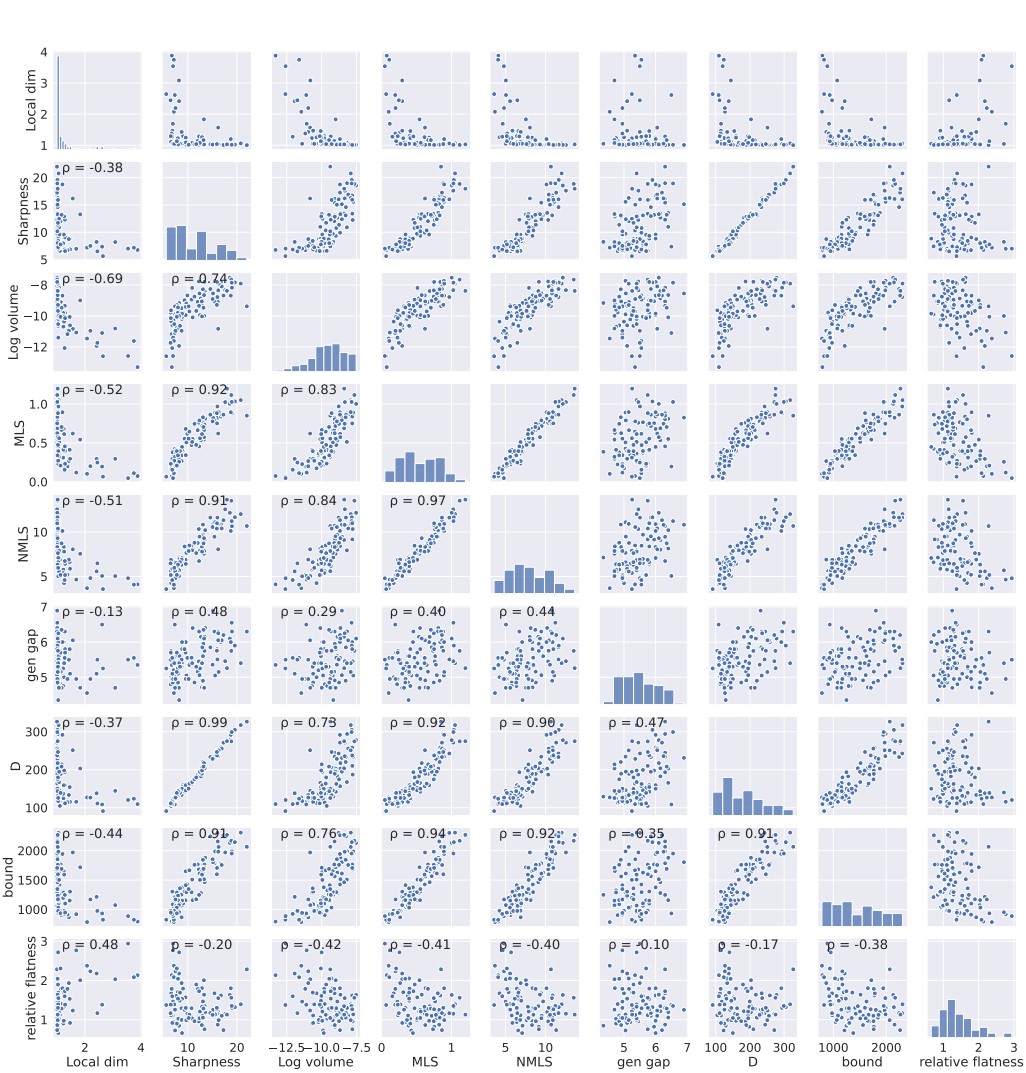

Figure G.5: **Pairwise correlation among different metrics.** We trained 100 different VGG-11 networks on the CIFAR-10 dataset using vanilla SGD with different learning rates, batch sizes, and random initializations and plot pairwise scatter plots between different quantities: local dimensionality, sharpness (square root of Equation (3)), log volume (Equation (7)), MLS (Equation (11)), NMLS (Equation (12)), generalization gap (gen gap), D (Equation (48)), bound (right-hand side of Equation (12)) and relative sharpness ((Petzka et al., 2021)). The Pearson correlation coefficient $\rho$ is shown in the top-left corner for each pair of quantities. See Appendix G.2 for a summary of the findings in this figure.

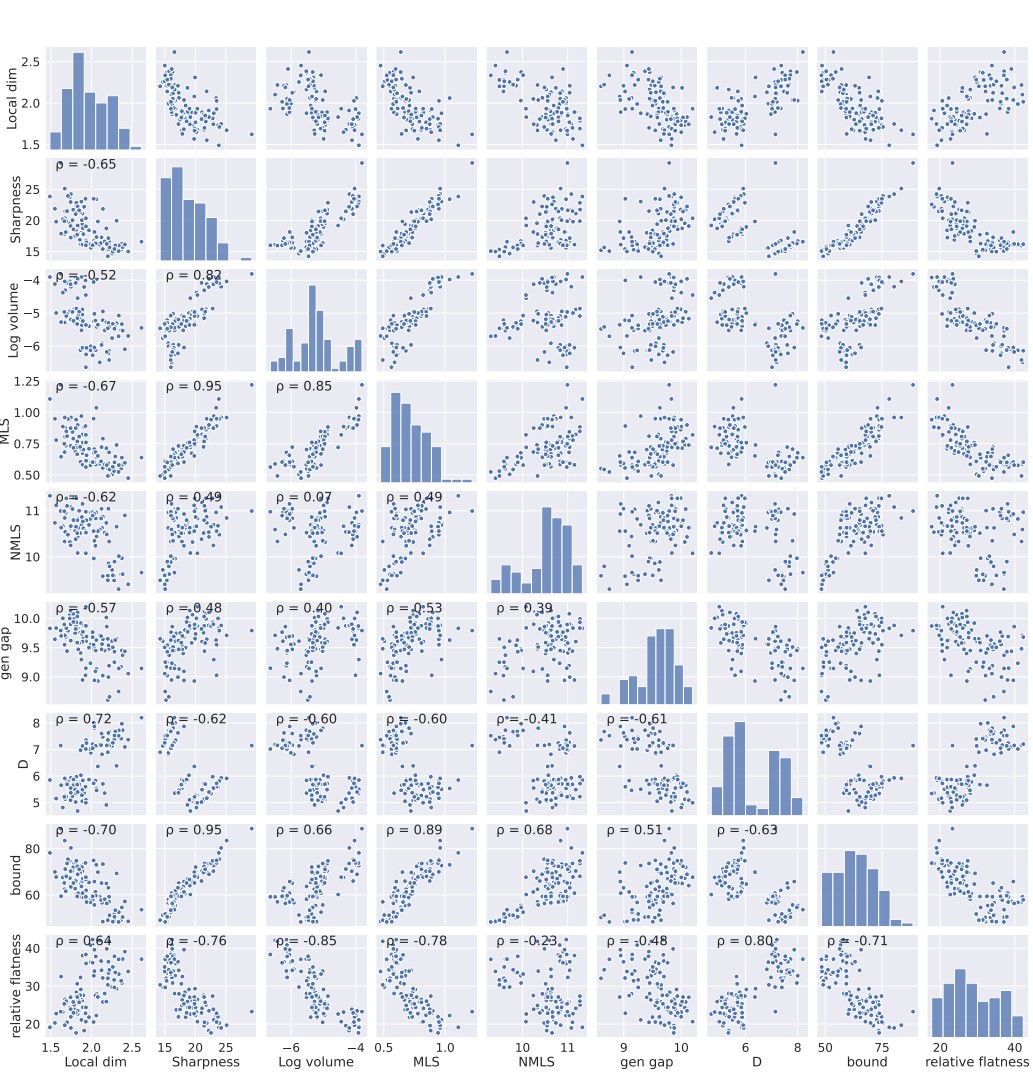

Figure G.6: **Pairwise correlation among different metrics.** We trained 100 different 4-layer MLPs on the FashionMNIST dataset using vanilla SGD with different learning rates, batch size, and random initializations and plot pairwise scatter plots between different quantities: local dimensionality, sharpness (square root of Equation (3)), log volume (Equation (7)), MLS (Equation (11)), NMLS (Equation (12)), generalization gap (gen gap), D (Equation (48)), bound (right-hand side of Equation (12)) and relative sharpness ((Petzka et al., 2021)). The Pearson correlation coefficient $\rho$ is shown in the top-left corner for each pair of quantities. See Appendix G.2 for a summary of the findings in this figure.

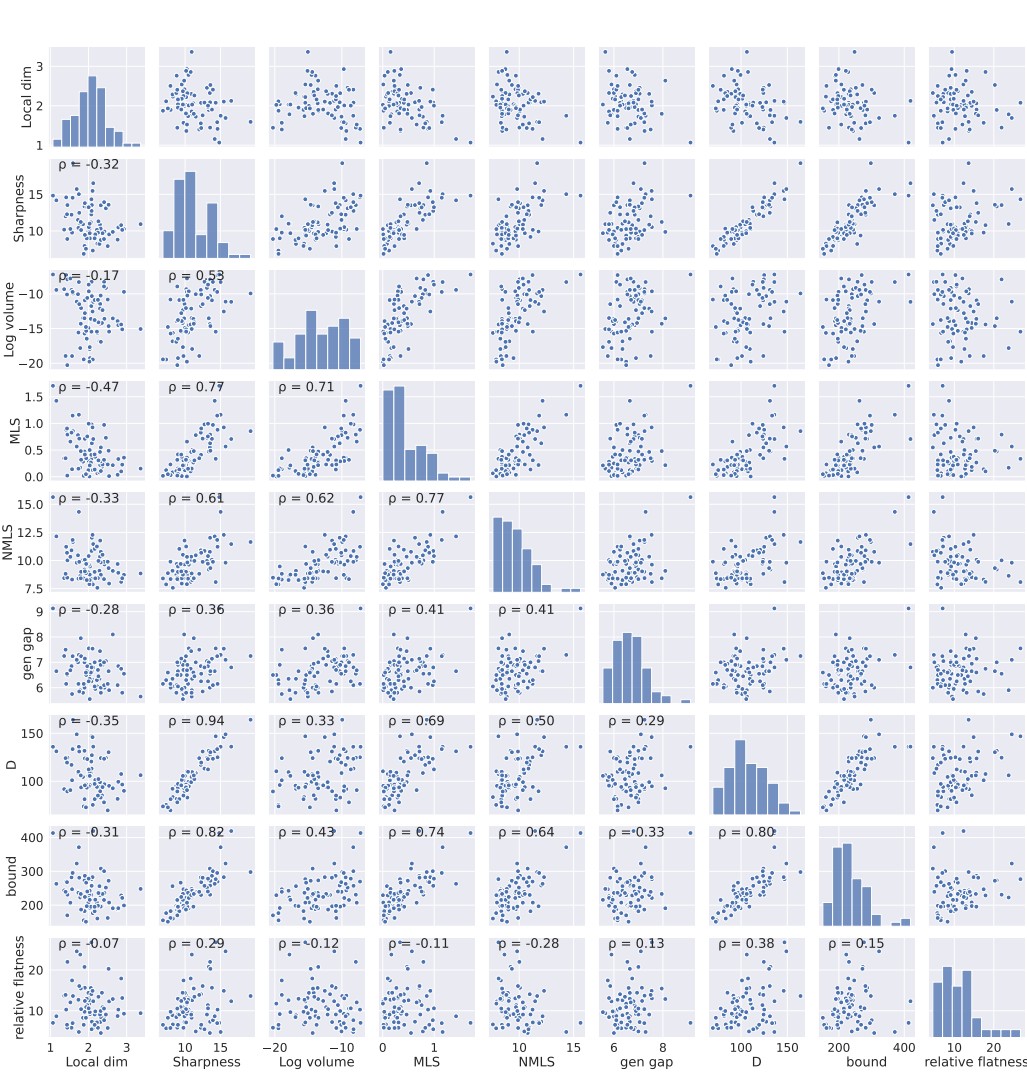

Figure G.7: **Pairwise correlation among different metrics.** We trained 100 different LeNets on the CIFAR-10 dataset using vanilla SGD with different learning rates, batch size, and random initializations and plot pairwise scatter plots between different quantities: local dimensionality, sharpness (square root of Equation (3)), log volume (Equation (7)), MLS (Equation (11)), NMLS (Equation (12)), generalization gap (gen gap), D (Equation (48)), bound (right-hand side of Equation (12)) and relative sharpness ((Petzka et al., 2021)). The Pearson correlation coefficient $\rho$ is shown in the top-left corner for each pair of quantities. See Appendix G.2 for a summary of the findings in this figure.

## H ADDITIONAL EXPERIMENTS

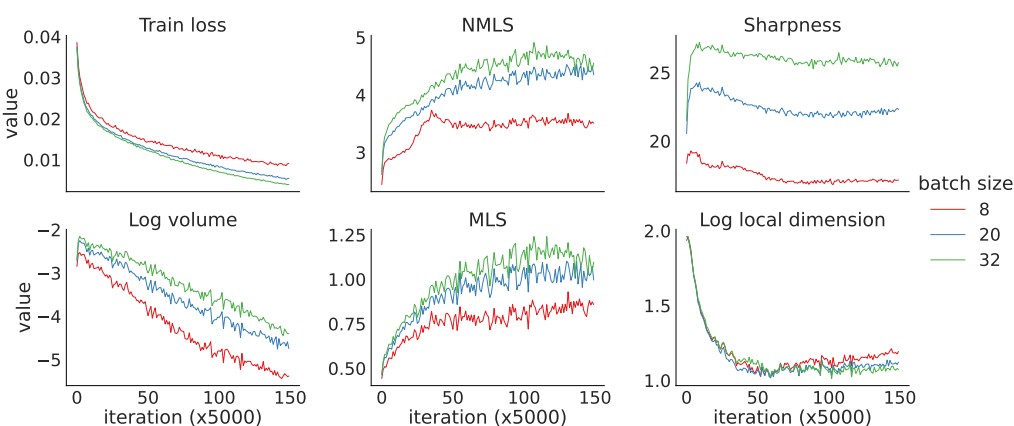

Figure H.8: Trends in key variables across SGD training of a 4-layer MLP with fixed learning rate (equal to 0.1) and varying batch size (8, 20, and 32). After minimizing the loss, lower batch sizes lead to lower sharpness and stronger compression. Moreover, MLS/NMLS closely follows the trend of sharpness during the training. From left to right: train loss, NMLS, sharpness (square root of Equation (3)), log volumetric ratio (Equation (7)), MLS (Equation (11)), and local dimensionality of the network output (Equation (15)).

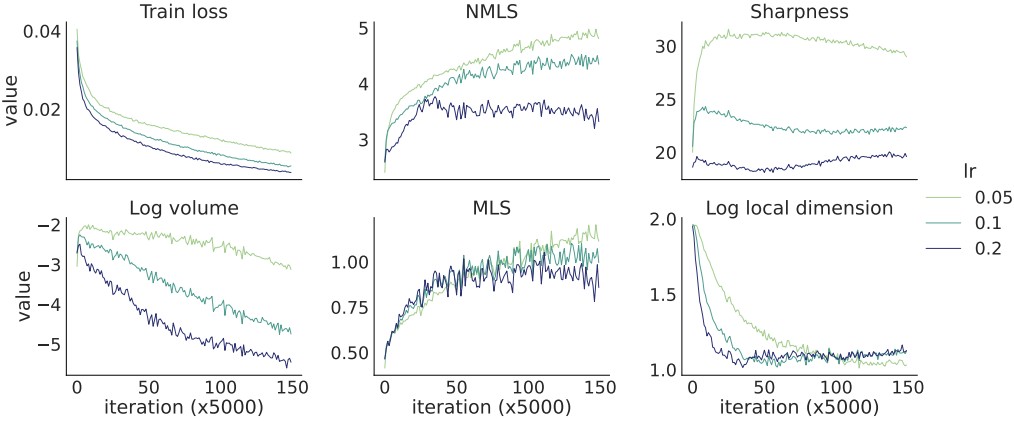

Figure H.9: Trends in key variables across SGD training of a 4-layer MLP with fixed batch size (equal to 20) and varying learning rates (0.05, 0.1 and 0.2). After the loss is minimized, higher learning rates lead to lower sharpness and hence stronger compression. Moreover, MLS/NMLS closely follows the trend of sharpness during the training. From left to right: train loss, NMLS, sharpness (square root of Equation (3)), log volumetric ratio (Equation (7)), MLS (Equation (11)), and local dimensionality of the network output (Equation (15)).

## I COMPUTATIONAL RESOURCES

All experiments can be run on one NVIDIA Quadro RTX 6000 GPU.