# OpenReview forum: "A simple connection from loss flatness to compressed representations in neural networks"
_ICLR.cc/2025/Conference — Submitted to ICLR 2025_

### Official Review · Reviewer_hd6L · 2024-10-31

**Soundness:** 3
**Presentation:** 3
**Contribution:** 3
**Rating:** 6
**Confidence:** 3

**Summary:**

The paper studies the connection between the sharpness of the loss landscape in the weight space (defined by the sum of eigenvalues of the Hessian matrix of loss wrt to weight, Eq. 2) and the local compression, or input sensitivity, in the feature space through several different measurements including "Local Volumetric Ratio", "Maximum Local Sensitivity", "Local Dimensionality". Building on the work of "Ma & Ying, 2021, On linear stability of SGD and input-smoothness of neural networks", the paper showed improvement on the previous theoretical bound between sharpness and input sensitivity (Eq. 6), extended the bound from input sensitivity of the first raw input to include all the intermediate representations across layers, and showed the empirical correlation between sharpness and input sensitivity in VGG-11 and MLP in CIFAR-10 and Fashion-MNIST dataset. The main bound improvement comes from replacing the highly variable scaling factor $||W||^k_2 / min_i ||x_i||^k_2$ (in Eq. 6) by tighter and more explicit scaling factors in Eq. 9 for Local Volumetric Ratio and Eq. 11 for Maximum Local Sensitivity.

**Strengths:**

Overall, the paper is well-motivated (addressed an existing problem), well-written (sufficient background and clear derivation of the theoretical results), and the results are based on sufficient theoretical ground.

**Originality**
1. The paper improves the previous theoretical results by *Ma & Ying, 2021* to introduce a tighter bound to relate input sensitivity in the feature space with sharpness in the weight space.
2. The paper introduces two new measurements "Local Volumetric Ratio (LVR)" and "Maximum Local Sensitivity (MLS)" which has tighter bound than the previous bound.
3. The author extend the input sensitivity equation (Eq. 10 and Eq. 12) to capture the sensitivity of not only the raw input but also the intermediate representation of all layers, therefore tightening the bound.
4. The author made an argument about why a previously introduced metric, "Local Dimensionality" is not bounded by sharpness.

**Quality**
1. The proofs related to the initial claim of how the new metrics "Local Volumetric Ratio" and "Maximum Local Sensitivity" can tighten the previous bound is presented in the paper, with detailed proof linked in the Appendix.
2. The theoretical claim is validated in practical dataset in VGG-11 with CIFAR-10 and MLP with Fashion-MNIST.

**Clarity**
1. The paper is well-written, with sufficient introduction of background and previous work, and careful explanation and acknowledgement of *Ma & Ying, 2021*, the main theoretical work that the paper is built on.
2. The technical definition of relevant terms (sharpness, volumetric ratio, local sensitivity, local dimensionality) is well-defined using precise notations.
3. The theoretical derivation of the main results (Eq. 5, Eq. 8, Eq. 11) is well-annotated, including which inequality and equality are used to arrive to the results, making the derivation is quite easy to follow.

**Significance**
1. The paper does address on the problem of previous loose bound and inconsistency in the correlation of sharpness and different input sensitivity methods.

**Weaknesses:**

While the paper is well-written with sufficient theoretical results, the empirical experiments as well as the practical significance of the paper can be improved.

**Quality**
1. While the main claim of the paper is that LVR and MLS tighten the bound, and therefore improve the correlation between sharpness and input sensitivity, the empirical results does not quantify the correlation between each current measurements (LVR and MLS), or the previous bound from *Ma & Ying, 2021* with sharpness. While visual inspection seems that they're more correlated, quantifying this correlation would make the finding stronger and more convincing.
2. [L373] The paper mentioned that the VGG-11 results was obtained on only 2/10 classes in CIFAR-10 so that the training time is faster. While limited computing resources is understandable, CIFAR-10 is already quite a small dataset and only take around several hours to converge on the whole dataset. Furthermore, training on 2 classes (binary choices) is a significantly easier task in 10 classes, which makes it difficult to be certain whether the theoretical results can be hold for a practical multiple category classification beyond binary classification. Therefore, the empirical results would be much more convincing if the authors can demonstrate that the theoretical results hold for the entire CIFAR-10 dataset and not only 2 classes.
3. Reproducibility: I couldn't find anywhere in the paper whether the authors mention they will release the code to reproduce the paper.

**Significance**
1. While the paper does address an existing problem, which is to improve the loose bound in *Ma & Ying, 2021*, the paper offers little insights on how these metrics can be further used to improve the generalizability or robustness of training machine learning models. One potential area to be mentioned is robust to common corruption, or to adversarial robustness, which is highly relevant to the input sensitivity mentioned by the paper. Also, the paper does show that with different hyper-parameters (learning rate, batch size), training loss all converge to zero, but sharpness and input sensitivity does differ, so there might be additional room here to use these metrics to study how different training hyper-parameters impact input sensitivity and sharpness, therefore offer some insights into model generalizability and robustness.

**Questions:**

**Questions**
1. [L250] Maybe I miss something, but I don't see why the equality in Eq. 8 does not hold in-practice? Why we need “would need all singular values of the Jacobian matrix $\nabla_x f$ to be identical?” Some additional clarification would be very appreciated.

**Suggestion**
1. [L172] Contribution section: Hyperlink ref the specific section that related to the improved bound to make it easier to follow. For example, the author mentions that they used geometric mean to improve the bound, and extend the previous results from scalar output to vector output, but it's not clear throughout the paper where these improvements are made.
2. Throughout the paper, the authors repeatedly use “representation’s compression”, which can be interpreted as both "local" or "global". Since the results mostly focus on "input sensitivity", or $|| \nabla_x f(x,\theta) ||$, the authors may consider use “local representation’s compression” or “robust to input perturbation” instead to avoid confusion for other metrics that measure more global scale of representation compression.

---

> ### Author Response · Authors · 2024-11-24
>
> Thank you for your review and analysis of our paper.
>
> Response to weaknesses:
> Quality:
> 1. In Appendix G, we perform the relevant calculation.  In our revision we will more explicitly bring this out in the main paper.
> 2. While this is a fair point for this experiment, we note that we do use 10 categories for other datasets we consider including FashionMNIST and CIFAR-100 in Figure 3.
> 3.  An important point: code will be released after acceptance.
>
> Significance:
> 1. We appreciate this point.  In our revision we will clarify that our core contribution is to elucidate the relationship between compression and sharpness, so as to understand the nature of sharpness and the recent sometimes contradictory results on the relationship between sharpness and generalization. Indeed recent work (Andriushchenko et al., 2023) and our results (figures in appendix G) both show that there may not be a uniform relationship between either sharpness or compression and generalization (please see also our response to the first reviewer nNBj.) Regarding the hyperparameters, both figure 1 and 2 and additional result in appendix H explores how learning rate and batch size affects compression. Due to space and the focus of our paper, we did not show the test loss, but smaller batch size and larger learning rate implies smaller test loss (we decide against showing this specific result here as the trend, as the reviewer likely knows, is well established (Jastrzębski, 2018)).
>
> Response to questions:
> 1. Empirically log volume keeps decreasing but MLS plateaus in Figure 1. Recall that volume is the product of eigenvalues and MLS is the largest eigenvalue. This implies that the eigenspectrum of C^{\lim}_f is becoming increasingly non-uniform, and only when all singular values of the Jacobian are uniform, can the equality in lemma 3.2 hold.
>
> Thank you also for the suggestions regarding presentation and phrasing in our paper.  We will adopt improvements along both of these lines in our revised version.

---

> > ### Comment · Reviewer_hd6L · 2024-11-26
> > **Response**
> >
> > Thank the authors for responding to my questions! Since the response haven't addressed the concerns to the lack of comprehensive empirical evaluation as well as the applicability of the proposed metrics to generalizability or robustness (more details mentioned above), I've decided to keep my initial score of "6: marginally above the acceptance threshold".

---

> > > ### Author Response · Authors · 2024-11-28
> > >
> > > Thank you for the discussion, and the decision to keep the score.
> > >
> > > We would like to reiterate that we consider all classes for  FashionMNIST and CIFAR-100 in Figure 3. The reason that we only consider 2-classes is the same as Ma & Ying, 2021, which is to make the train loss quickly converge to 0, so that our theoretical assumption is quickly met. Therefore, combined with results on FashionMNIST and CIFAR-100 with different network architectures, we deem our empirical results to be sufficient.
> > >
> > > We also kindly remind the reviewer that this is a theoretical paper that aims to understand the nature of sharpness and compression, and have similar practical implications to Andriushchenko et al., 2023, which challenges the relationship between sharpness and generalization. By showing the close connection between sharpness and compression, we gain deeper understanding of sharpness and how minimizing it contributes to compression/robustness. The matter of how to minimize sharpness is handled by a separate body of literature that includes SAM (Foret et al., 2020) and Jastrzębski, 2018.

---

### Official Review · Reviewer_AUpH · 2024-11-03

**Soundness:** 3
**Presentation:** 3
**Contribution:** 3
**Rating:** 6
**Confidence:** 4

**Summary:**

This paper explores the connection between sharpness in the loss landscape of neural networks and the geometric compression of neural representations. By examining sharpness and compression, the authors present a dual perspective bridging the parameters and feature space. They introduce two main metrics for quantifying representation compression: Volumetric Ratio and Maximum Local Sensitivity (MLS). These metrics provide a theoretical basis to show that flatter minima in the loss landscape imply a certain degree of representation compression. The authors conduct empirical experiments on multiple architectures, such as VGG-11 and MLP, to verify their theory. They demonstrate a strong correlation between representation compression and sharpness, which suggests implications for the generalization capacity of neural networks.

**Strengths:**

1. The authors provide a well-reasoned approach linking sharpness in parameter space with compression in feature space. This connection between sharpness and compression is both mathematically rigorous and conceptually nice.
2. The paper effectively combines theoretical findings with empirical results, using varied architectures (e.g., VGG-11, LeNet, ViT) to showcase the consistency of their bounds across different network structures.

**Weaknesses:**

1. Local Dimensionality: In Section 3.3, the theoretical treatment of local dimensionality lacks explicit bounds, making it difficult to evaluate its practical impact. While related arguments, such as those relying on Taylor expansion, are given, a formal bound for local dimensionality would strengthen the contribution.
2. Indirect Connection to Generalization: Although the paper is motivated by generalization, the theory largely focuses on sharpness and compression without direct results or proof of their causal link to improved generalization capacity. This could lead to a misinterpretation of the paper's goals, as the title and introduction emphasize generalization.
3. Presentation of Results: Certain results, particularly those related to local dimensionality (Figure 1 and 2), suggest that local dimensionality increases during training, which appears to contradict the central claim of compression. This inconsistency could be addressed by clarifying the behavior of local dimensionality in the final stages of training.

**Questions:**

1. Definition of C_f^{\lim}:Could you clarify the definition of C_f^{\lim} (line 204)? It would be helpful to refer to the appendix and add some intuition in the main text.
2. Relation to Neural Tangent Kernel (NTK): Have you considered potential connections between your sharpness metric (Eq. 3, line 130) and the Neural Tangent Kernel? This would be particularly interesting for understanding if flat solutions introduce certain inductive biases through gradient flow.
3. Generalization Motivation: The introduction appears to frame the paper around generalization, yet the theoretical focus is on sharpness and compression. Could you clarify whether the link to generalization is correlational or whether there are theoretical underpinnings that relate compression directly to generalization?

---

> ### Author Response · Authors · 2024-11-24
>
> Thank you for your close reading, care, and time in preparing this review.
>
> Response to weaknesses:
>
> Point 1 and 3: We explain the fundamental difference between local dimensionality and other compression metrics (volume and MLS) in section 4.2. That is, the local dimensionality measures how uniform the eigenspectrum is, while volume and MLS measures the magnitude. Therefore it is expected that local dimensionality does not correlate well with either compression or sharpness. We do show, however, there is a notable correlation between compression and sharpness.  We believe that presenting and explaining this range of results in fact strengthens our paper, but appreciate that we need to bring out the distinctions and the underlying reasons for them more clearly in our paper, and will do so in our revision.
>
> Point 2: We appreciate this point, and will rework our narrative, especially in the introduction, to make clear our results, including the valid point of the reviewer that connections to generalization are indirect.  We will clarify that our core contribution is to elucidate the relationship between compression and sharpness, so as to understand the nature of sharpness and the recent contradictory results on the relationship between sharpness and generalization. Please see also our response to the first reviewer nNBj.
>
> Response to questions:
> 1. C_f^{\lim} is defined in equation 14, and we will explicitly note this in the revision.
> 2. This sounds interesting -- we have not yet, but it seems an interesting direction for future work -- we would be grateful to know if there are specific citations that could form a springboard for this and/or that would be appropriate to include at this point.
> 3. Thank you for asking us to clarify.  As the reviewer rightly implies, indeed recent work (Andriushchenko et al., 2023) and our results (figures in appendix G) both show that there may not be a uniform relationship between either sharpness or compression and generalization. While we note this in the introduction (line 46-55), we agree that our narrative should be reworked to make it clear that, as the reviewer rightly states, our main focus is on the relationship between sharpness and compression.

---

### Official Review · Reviewer_FRnE · 2024-11-07

**Soundness:** 3
**Presentation:** 3
**Contribution:** 2
**Rating:** 5
**Confidence:** 3

**Summary:**

The paper establishes a new connection between the flatness of neural network loss landscapes and the compression of neural representations, bridging two avenues that at first glance seem fairly orthogonal for gaining insights into generalization.
The authors derive a mathematical relationship which shows flatter loss landscapes are connected to tighter upper bounds on certain compression metrics, namely volumetric ratio and maximum local sensitivity (MLS).

There are three primary contributions spread throughout this work. The approach introduces explicit bounds linking loss landscape sharpness to neural representation compression through two core metrics: volumetric ratio and MLS. Empirical experiments on several architectures of neural networks, including VGG-11, LeNet, MLP, and ViT, drive it to acquisition of direct evidence of the strong correlation between sharpness and representation compression. Finally, the authors demonstrate in Proposition 3.10 that adding all the linear weights of the network leads to bounds that reliably predict positive correlations between compression and sharpness across a range of experimental conditions.

In summary, this work provides an alternate view of understanding generalization for neural networks, in which both the representation space of compression and the parameter space of sharpness are potential conduits through which great insights into the generalization behavior of a trained model can be found.

**Strengths:**

The novelty of this paper is bridging the gap between loss landscape sharpness and neural representation compression, two concepts that have been tackled separately from one another as regards neural generalization throughout. In other words, this paper lets researchers think about generalization in a different way in both the parameter and the feature space with explicit mathematical bounds explicitly linking them. It is not a formulation of another variant of ideas already existing but an exciting mixture of theoretical insight and empirical evidence which extends our knowledge of neural network behavior during training conditions.

The authors are technically intense on the quality of derivation of inequalities and bounds, theoretically grounded in known concepts such as sharpness and neural representation metrics. Also, a wide range of experiments performed for multiple architectures include VGG-11, LeNet, MLP, and ViT that justify their claims to validate the proposed bounds hold consistently. Hence, this empirical evidence, together with theoretical clarity, shows robustness in the claims by the paper, increasing its reliability.

Overall clarity is high. The authors have provided various obvious reasons why it becomes valuable and necessary to establish a link between loss sharpness and compression of representation by designing and explaining complex concepts in logical sequence.

There is an excellent organization of mathematical derivations and the relationship of theory to experimental results. Such illustrative figures and examples presented in the experiments section will help readers perceive the practical implications of theoretical results. However, some readers might find parts of the theoretical development a little too dense.

The paper could be of significant impact in relation to the understanding of neural network generalization. The paper establishes new pathways for future research by providing a framework connecting sharpness in loss landscapes to compression in neural representations. While this contribution itself closes an elementary gap in the literature, the tools and insights developed will likely impact a very wide range of applications in which generalization is a critical challenge. The combination of theoretical depth and practical implications would make this work worthwhile to all researchers and practitioners interested in factors that contribute to neural network robustness and efficiency. On a balance, the paper constitutes a meaningful contribution to the discipline and brings into focus a stage for further investigation of dual perspectives in network analysis.

**Weaknesses:**

Although a very good case is built on linking loss sharpness to representation compression, one could notice improvement in expansion of the theoretical framework and more empirical evidence to back the results with clarity in parts of the paper.

In theory justification for using compressed metrics like volumetric ratio and MLS are notably improved. Despite these measurements proposed and being bound, had the authors provided further clarification why these particular measurements better describe the phenomenon of compression in feature space, a tightened contribution would have been experienced. For instance, comparisons with possible compression or robustness measures such as different sensitivity measurements or mutual information-based measurements may perhaps provide more context about their choices and therefore make it clearer what significance the work is prone to have. This would also help to address possible shortcomings of generalizability, especially for architectures beyond those tested.

The experimental study, being wide-ranging, can actually usefully be augmented by further settings and comparative baselines. While the study spans multiple architectures and datasets, extending experiments to consider other tasks or domain-specific architectures, such as natural language processing or reinforcement learning, would further enhance the robustness of results by demonstrating that these correlations between sharpness and compression were generalizable across vision-oriented models primarily used in experiments. Additional baselines to compare, including networks trained with adaptation techniques, such as Adam, in addition to explicit regularization techniques, might provide deeper insights into the generality of these sharpness-compression relationships between different paradigms of training.

Some parts of the text are also unclear, and more room can be improved on some derivations of the bounds. Some of the derivations, such as those involved in the proofs of some versions of Propositions 3.4 and 3.10, are mathematically quite involved and might become clearer with more explanatory text or with simplified examples which make transparent the intuition behind the conclusion reached at each place. Including more visuals, such as either diagrams or graphs, which depict these mathematical relationships could help a wider audience comprehend these theoretical contributions, especially for practitioners who would be unfamiliar with the nuances of such bounds.

Finally, the work is meaningful in advancing a dual perspective on the generalization of neural networks but could really drive it home with some practical implications on exactly how those identified relationships between sharpness and compression may be actually leveraged in model selection or designing training regimes that positively affect generalization. For instance, it would be great to discuss whether these bounds could inform hyperparameter choices, learning rate schedules, or model architectures chosen.

In a nutshell, this paper stands as a strong and valuable contribution, only that an expanded theoretical comparison, diversified experimental setting, enhanced clarity in the presentation of derivations, and practical insights on how its findings could influence model training and evaluation in real world could make the impact much greater.

**Questions:**

1. Could the authors elaborate why volumetric ratio and MLS were chosen as the primary metrics for compression? For instance, could you further explain why these metrics are better suited to capture the representation compression related to sharpness? This would be aided by adding discussion over whether alternative compression metrics such as mutual information or other sensitivity measures would go on to outline the choices made here and potentially elevate the theoretical contribution.

2. While the experiments cover a wide variety of architectures, they are still highly skewed toward vision-based models and tasks. How might you discuss extending this relationship to another domain, such as NLP or reinforcement learning? How might you consider making extensions towards models or tasks in these areas to further demonstrate applicability?

3. The empirical results show that there is an interaction between sharpness and compression for standard SGD-trained models. Would the authors be interested in checking if this interaction is also seen for models optimized with adaptive techniques like Adam or RMSprop, or even with other explicit regularization techniques, such as dropout or weight decay? Anything concerning their possible impact on the achieved sharpness-compression bounds can help further assess the strength of the conducted claims.

4. Some of the theoretical chapters, for example those with bound derivations (such as Propositions 3.4 and 3.10), are mathematically very dense. Some extra examples and diagrams would help in getting these ideas across visually as one reads through the theorem and its derivative.

5. Here, the authors claim that the learning rate and the batch size do affect the sharpness, which consequently affects compression. Do the authors have some more elaborate analysis about how certain choices of the hyperparameters impact these measures? Are there certain selections of hyperparameters which improve the relationship between sharpness and compression or would degrade it? This would relate the results even more practically to training models and the choice of the hyperparameters during practice.

6. What are the practical implications of your findings in terms of model selection and training? While, as a theory paper, the discussions have naturally led to theoretical insights, it would be useful to comment on practical implications drawn from the findings on model training and selection. For instance, how might the relationships identified between sharpness and compression inform choices in architectures, training regimes, or model selection? Guidance on how to leverage such insights in practical settings would strengthen the impact of the findings on practitioners.

7. The authors consider only the later stage of training where the models are likely within the interpolation regime. Could more context or empirical evidence be given on how compression metrics evolve over earlier phases of training? This could help to explain if and how the relationship between sharpness and compression evolves over training and sheds additional light on how relevant these results are to generalization throughout that entire training process.

---

### Official Review · Reviewer_dXon · 2024-11-07

**Soundness:** 2
**Presentation:** 2
**Contribution:** 2
**Rating:** 5
**Confidence:** 3

**Summary:**

This paper aims to connect the sharpness of the loss surface in parameter space with the compression of inputs in feature space. The Network Volumetric Ratio (NVR) and the (Network) Maximum Local Sensitivity (NMLS) are proposed for measuring compression in terms of volume compression and sensitivity to input changes, respectively. These measures are upper bounded by expressions that include sharpness as measured by the Hessian trace, for models that interpolate the training data. Experiments mostly show positive correlation between NVR, (N)MLS, their upper bounds, sharpness and generalization.

**Strengths:**

The paper tries to make the bound of input gradients by parameter gradients by Ma and Ying (2021) more informative by employing versions of the linear stability trick to various proposed metrics of compression. The authors explain this background well and propose a series of metrics to correct weaknesses of existing upper bounds of compression through sharpness. Their bounds consider all trainable weights instead of just the first layer, replace the minimum over training points by their geometric mean, and consider the maximal singular value instead of their product to correct the weakness of this paper’s volumetric notion of compression. Limitations concerning connections between sharpness and generalization, or of the proposed compression metrics and bounds are partially acknowledged and investigated.

**Weaknesses:**

Overall, I am leaning toward rejection, primarily because the connections to generalization - while heavily emphasized in the story - remain underexplored, the limited amount of evidence remains inconclusive and the presentation, in particular starting from Section 3.4, could be greatly improved.

- **Implications on generalization underexplored.** While the paper’s narrative heavily refers to generalization as an important ultimate measure of interest for studying sharpness and compression, neither the theoretical nor the empirical results provide a sufficient connection to generalization.
    - More theoretical and/or empirical evidence about connections to generalization would have been necessary. For example in Figure 3, you could include generalization by scaling the marker size by e.g. their generalization error quantiles. This would yield valuable information: Does generalization correlate well with MLS? Does it only correlate well within model classes?
    - The proposed measures of compression are not immediately connected to generalization as none of them measures whether the learned features are meaningful. Volume compression or small input gradient alone do not imply meaningful compression. It remains completely unclear whether the most informative or best-generalizing features are kept, and insights into causal connections are not studied. The limitation that sensitivity to input changes is sometimes desired for good generalization is acknowledged in the ‘needle in the haystack’ example.
- **Correlations remain inconclusive.** The existence of double descent has shown the danger of drawing conclusions from upper bounds that may sometimes be informative, but are often much too loose. The theoretical bounds are immediate and so generic that they have to be loose in many cases. Even the correlation between NMLS and sharpness is less clear for MLPs in Figure H.9 for which the presented upper bound applies. The correlation to generalization provided in Appendix G.2 is even less pronounced. The caption of Figure G.4 even acknowledges a negative correlation between the left and right hand sides of eq. (11).
- **Limited empirical evidence.** The provided Figures 1 and 2 seem anecdotal, only considering a few training runs of one model on one data set. The slightly more systematic correlation figures in Appendix G.2 present inconclusive results. The correlations are weak and strongly vary across the tested model and dataset combinations. Comparable studies like Andriushchenko et al. (2023) and Jiang et al. (2019) that just report correlations of measures with generalization are much more extensive in terms of settings, generalization measures and models tested.

**Minor criticism concerning the presentation:**

- The sentence between l. 350-353 does not contain a verb.
- Normalized sharpness in Figure 3 comes out of the blue. Its connection to elementwise adaptive sharpness should be explained in the main text.
- The notation Vol(x) in eq. (7) is undefined. A priori, a point does not have any volume.
- In l. 299, calling a Gaussian random variable ‘ball’ is unconventional and at least requires a justification. ‘depends on’ should rather be ‘determines’
- In Appendix G.2, the terms 'D' and 'bound' are confusing. I suggest finding better abbreviations and stating that 'D' refers to the right-hand side of the MLS bound and 'bound' to the right-hand side of the NMLS bound. Also I would evaluate the adaptive sharpness and compression measures from Appendix E here.

**Typos:** 163, 264: first, 280, 363: small t in the minimum?, 379-380: several mistakes.

**Questions:**

1. Why are you not introducing and evaluating reparameterization-invariant measures of sharpness and the related compression bounds more thoroughly even in the main body, if they seem to be more sensible than the naive Hessian trace?
2. Replacing the product of singular values of the Jacobian $\nabla_x f$ by the maximal singular value seems to be the first natural idea for spread out spectra, but also seems to provide very limited information into the properties of $\nabla_x f$ (only one-dimensional information). So do you think there will be more refined and more useful measures in the future or is the maximal singular value enough?
3. Have you thought about other measures of compression that relate to generalization more immediately?
4. Why does NMLS generally seem to be less correlated with the other quantities than MLS in Appendix G.2?

---

> ### Author Response · Authors · 2024-11-24
>
> Thank you for your careful analysis and all of your time in preparing this review.
>
>
> Response to weakness:
> - Implications on generalization underexplored: We appreciate this point, and see that our narrative, especially in the introduction, gave a misleading impression of what we will accomplish with respect to generalization.  The reviewer is absolutely correct that our work does not produce a uniform connection between compression (or flatness) and generalization, and that our links to generalization overall are correlative when they are made.  We do believe that this is a real consequence of the underlying complexity:   the work of others (Andriushchenko et al., 2023) as well as some of our own findings (Figure G.5-7) challenge the existence of a systematic link between sharpness/compression and generalization across multiple models and settings.  Rather, the contributions of our paper are to link two quantities -- sharpness and compression -- that are often considered in separate treatments of generalization that have come to widely varied conclusions across the literature.   We will change our paper in two ways to improve it in this regard.  First, we will the writing of our narrative much more much more clear on this point in our revision and again thank the reviewer for encouraging us to do so.  Second, we will make changes to the figures, such as that suggested by the reviewer with respect to marker sizes, to better bring out key points regarding generalization.  Please also see our response to point 1 of reviewer nNBj.
>
> - Correlations remain inconclusive: “The existence of double descent has shown the danger of drawing conclusions from upper bounds that may sometimes be informative, but are often much too loose.”:  We definitely appreciate this point about upper bounds, and this concern motivates much of the analysis in our paper.  For example, we were concerned that the original theoretical bound proposed in Ma & Ying, 2021 would not be able to explain the correlation between sharpness and MLS, and this was indeed the case, as demonstrated in Figure G.4 with the negative correlation between the left and right hand sides of Eq. 11 (please note that Eq. 11  is a slightly improved version of the inequality derived in Ma & Ying, 2021 (see also line 172-175)).  Our NMLS metric, on the other hand, maintains positive correlation with the other side of the inequality in all cases, demonstrating its utility.  We explain the reason after Proposition 3.10: because now all network weights appear on both sides of the inequality (see also line 899-900 in the proof).  This said we agree with this overall concern and in our revision we will both state it more clearly at the outset of the paper, and will more explicitly note how our results are both limited by it and, for cases such as that specified above, are able to make progress.
> - Limited empirical evidence: We note that each line in Figure 1 and 2 is averaged across 5 random seeds. While we could not present all empirical results in the main text fully, we note that there are additional empirical results are in the appendix. However, respectfully note that we believe that even Figure 1 and 2 show that MLS and NMLS are well correlated with sharpness, while NMLS is even more correlated.
>
> Questions:
> 1. The Hessian trace is the most used definition by the other papers, so we start from it and show that extension to reparametrization-invariant measures is possible.
> 2.  We observe that the distribution of singular values quickly decays after the largest singular value, so we believe that most information is preserved in the largest singular value.
> 3.  Thank you for this suggestion -- one set of related measures that we are aware of here are those involved in the information bottleneck work.  We will add citation to this work in our revision, in the context of the possible advances that the reviewer suggests.  However, as this work is at least classically phrased in terms of overall conditioned entropy, rather than the more explicit and localized geometrical terms we use here, we believe that a direct link to the bounds we use here will require significant additional theoretical extensions.  We will additionally clarify and biring out our discussion of allied measures, around line 400-405, and the challenges they may bring.
> 4. This might be a case where reparameterization-invariant measures become especially important. In fact, in figure 3, if normalized MLS is not used, no correlation will be observed. As an indication, NMLS is better correlated with its bound in proposition 3.10.

---

> > ### Comment · Reviewer_dXon · 2024-11-29
> >
> > I thank the authors for their response to my criticism and questions. Since my main concerns remain, I will keep my score leaning toward rejection. But I believe that this paper has the potential to be broadly well-received and accepted after a thorough revision. I recommend a more precise narrative, in particular a closer inspection of implications on generalization (as acknowledged in the rebuttal), a larger focus on reparameterization-invariant sharpness, more reliable empirical evidence and potentially a reconsideration of compression metrics with stronger connections to generalization.

---

### Official Review · Reviewer_nNBj · 2024-11-08

**Soundness:** 2
**Presentation:** 1
**Contribution:** 2
**Rating:** 3
**Confidence:** 3

**Summary:**

The authors propose 3 metrics of representation compressibility in neural networks, two of which they can upper bound with a quantity related to the flatness of the solution in the parameter space by improving on a solution from previous work. They conduct experiments to empirically validate their proposed upper bounds.

**Strengths:**

- Establishing systematic connections between parameter properties (re. the loss landscape) and representation properties is a worthy and important effort.
- The properties the authors address (flatness of the minimum and repr. compression) are well chosen, as they are important notions for research on generalization in deep learning.
- The authors present interesting discussions at times (e.g. L400 - L405)

**Weaknesses:**

- I think the paper's constributions are unclear, they do not necessarily match how they are presented in the abstract and the introduction, and the paper is often silent or inconsistent as to implications of their findings. For example, the paper claims in the abstract that "Overall, we advance a dual perspective on generalization in neural networks in both parameter and feature space." Yet, the connection between the authors' proposed compression metrics and generalization is never explicitly discussed beyond some abstract discussion. In addition to not being connected to generalization in a principled way, the proposed metrics are not discussed in relation to existing compression/compressibility metrics in the literature. As a result, it's not clear for the reader why the bounds proposed by the paper should be important. In addition, how these metrics are discussed is also not consistent throughout the paper: they are first introduced as compression metrics, but after a point paper starts to discuss them as robustness-related notions, without any clear indication as to why. That these metrics empirically correlate with generalization gap is insufficient justification for these metrics as they are the core contribution of the paper.
- Connected to the point made above, the paper is not situated well in the context of existing literature. The paper's summary of literature on generalization sounds very limited, as it omits vast subsets of literature such as those that rely on model complexity or algorithmic stability. This is not limited to omitting semi-related subsets of the literature: e.g. the paper includes no explicit discussion of information bottleneck theory, which is a theory of generalization based on the compression of representations. Moreover, while mentioning the importance of representation properties in relation to generalization, the paper does not present a nearly comprehensive enough review of the literature, and suffices with the citation of a few papers on neural collapse.

**Questions:**

- L073: A better justification regarding why this part of the learning process is focused on would be useful
- L140: Please make it explicit what parts of the content between L140 and L160 is this paper's contribution and what parts are due to Ma & Ying (2021).
- L177: Are Ma & Ying's results related to scalar outputs or scalar inputs or outputs?
- L234: Is there a particular reason why sum is chosen as the aggregation method across layers?
- L370: What is representation collapse?

========

## Post-author feedback comments
I thank the authors for their response. Although I appreciate their willingness to modify the paper so as to better situate their contributions, I do not think the changes needed are manageable within one review cycle (even the abstract includes claims regarding generalization). Thus, I cannot recommend the paper's acceptance. That being said, I do think the line of inquiry adopted in the paper is potentially fruitful for deep learning theory, and encourage the authors to pursue this line of research further.

---

> ### Author Response · Authors · 2024-11-24
>
> Thank you for your care and time in preparing the review.
>
> Response to weakness: The reviewer is absolutely correct that our work does not produce a systematic theoretical link between compression (or flatness) and generalization, and that our links to generalization overall are correlative when they are made.  Indeed, the work of others (Andriushchenko et al., 2023) as well as some of our own findings (Figure G.5-7) challenge the existence of a systematic link between sharpness/compression and generalization across multiple models and settings.  Rather, the contributions of our paper are to link two quantities -- sharpness and compression -- that are often considered in separate treatments of generalization that have come to widely varied conclusions across the literature.  We understand how our writing in the first part of introduction gave a somewhat misleading impression of what we will accomplish with respect to generalization.  We will make this writing much more clear in our revision and appreciate the reviewer in prompting us to do so.
>
> In more detail, we will bring out the message of line 46-55: there has not been a definitive relationship between sharpness and generalization so far, and by understanding its close relationship to compression, we can propose a possible explanation for this weak relationship between sharpness and generalization ((L400 - L405), i.e. compression is not all you need).   As for the concept of compression, since both the volume and MLS are closely related to the singular values of the Jacobian, we consider them both as compression metrics. Intuitively MLS is the compression measured in the direction that explains most variance, which happens to also be the robustness of network output w.r.t. the input.
>
> Given the nature of our core contribution, literature on generalization that studies factors other than sharpness and compression such as those that rely on model complexity or algorithmic stability are beyond the core scope of our paper, but we will aim to cite one or several reviews on these broader topics.   We will cite in more detail the information bottleneck theory in our revision: thank you for this good point.
>
> Response to questions:
> - L073: Thank you, we will add a clarification where suggested: this is where the approximation of sharpness holds (appendix B).
> - L140: Our contribution is described in line 176-178. The equation (1) of Ma and Ying is also incorrect in its formula, so we helped to clarify both the statement of this equation and give a refined proof.
> - L177: For these results:  just scalar outputs.
> - L234: This aggregation method is chosen because it falls naturally from the operations involved in the derivation.
> - L370: We use this term in a manner analogous to compression.

---

### Meta-Review · Area_Chair_vXaX · 2024-12-20

**Metareview:**

This paper studies the connection between the sharpness of the loss surface in parameter space and the compression of representations in feature space. It introduces two criteria for measuring compression: the Network Volumetric Ratio (NVR), based on volume compression, and the (Network) Maximum Local Sensitivity (NMLS), based on sensitivity to input changes. For models that interpolate the training data, these measures are upper-bounded by expressions involving sharpness, as measured by the trace of the Hessian. This provides a theoretical basis demonstrating that flatter minima in the loss landscape imply a degree of representation compression. The authors conduct empirical experiments on various architectures, such as VGG-11 and MLP, to verify their theory and demonstrate a correlation between representation compression and sharpness.

Reviewers found the connection between loss landscape sharpness and neural representation compression to be important and interesting, praising the paper for its rigor and clarity. However, they also raised concerns, with most rating the paper as borderline and one recommending rejection. A common concern among reviewers nNBj, dXon, and AUpH was the inconsistency between the paper's claimed contributions and its actual results. They believe the narrative misleadingly emphasizes generalization (e.g., the quote from the paper stating "Overall, we advance a dual perspective on generalization in neural networks in both parameter and feature space"), while the actual results are on the connection between sharpness and compression. Although the authors acknowledged this in their response, admitting their contribution primarily focuses on compression and sharpness, the reviewers remained unsatisfied. They believe the narrative requires significant changes, or that additional results are needed to support the claimed effect of the proposed measures on generalization—either way necessitating a complete review cycle. While there were other insightful discussions, the issue with the presentation of contributions ultimately overshadowed these.

I do hope that authors are not discouraged by the reviews and the final recommendation, especially since both I and the reviewers find the paper's results interesting and useful. I encourage the authors to incorporate the feedback regarding the claimed contributions and resubmit their work.

**Additional Comments On Reviewer Discussion:**

Reviewers found the connection between loss landscape sharpness and neural representation compression to be important and interesting, praising the paper for its rigor and clarity. However, they also raised concerns, with most rating the paper as borderline and one recommending rejection. A common concern among reviewers nNBj, dXon, and AUpH was the inconsistency between the paper's claimed contributions and its actual results. They believe the narrative misleadingly emphasizes generalization (e.g., the quote from the paper stating "Overall, we advance a dual perspective on generalization in neural networks in both parameter and feature space"), while the actual results focus on the connection between sharpness and compression. Although the authors acknowledged this in their response, admitting their contribution primarily focuses on compression and sharpness, the reviewers remained unsatisfied. They believe the narrative requires significant changes, or that additional results are needed to support the claimed effect of the proposed measures on generalization—either way necessitating a complete review cycle. While there were other insightful discussions, the issue with the presentation of contributions ultimately overshadowed these.

---

### Decision · Program_Chairs · 2025-01-22

Reject